# Orographic and convective gravity waves above the Alps and Andes mountains during GPS radio occultation events – a case study

Rodrigo Hierro[1], Andrea K. Steiner[2,3], Alejandro de la Torre[1], Peter Alexander[4], Pablo Llamedo[1], Pablo Cremades[5]

*Correspondence to*: Rodrigo Hierro (rhierro@austral.edu.ar)

[1] Facultad de Ingeniería, Universidad Austral and CONICET, Pilar, Provincia de Buenos Aires
B1629ODT, Argentina

[2] Wegener Center for Climate and Global Change (WEGC), University of Graz, Graz, Austria

[3] Institute for Geophysics, Astrophysics, and Meteorology/Institute of Physics, University of Graz, Graz, Austria

[4] IFIBA, CONICET, Ciudad Universitaria, Buenos Aires, Argentina

[5] Facultad de Ciencias Exactas y Naturales,  Universidad Nacional de Cuyo, Argentina

**Abstract.** Gravity waves (GW) and convective systems play a fundamental role in atmospheric circulation, weather, and climate. Two usual main sources of GW are orographic effects triggering mountain waves and convective activity. In addition, GW generation by fronts and geostrophic
adjustment must also be considered. The utility of Global Positioning System (GPS) radio occultation (RO) observations for the detection of convective systems is tested. A collocation database between RO events and convective systems over sub-tropical to mid-latitude mountain regions, close to the Alps and Andes is built. From the observation of large amplitude GW structures under the absence of jets and fronts, subsets of RO profiles are sampled. A representative case study among those considered at each
region is selected and analyzed. The case studies are investigated using mesoscale WRF simulations, ERA-Interim reanalyses data and the measured RO temperature profiles. The absence of fronts or jets during both case studies reveals similar relevant GW features (main parameters, generation and propagation). Orographic and convective activity generates the observed GW. Mountain waves above the Alps reach higher altitudes than close to the Andes. In the Andes case, a critical layer prevents the
propagation of GW packets up to stratospheric heights. The case studies are selected also because they illustrate how the observational window for GW observations through RO profiles admits a misleading interpretation of structures at different altitude ranges. From recent results, the distortion introduced in the measured atmospheric vertical wavelengths by one of the RO events is discussed as an illustration. In the analysis, both the elevation angle of the sounding path (line of tangent points) and the gravity wave aspect

ratio estimated from the simulations and the line of sight are taken into account. In both case studies, a considerable distortion, over- and underestimation of the vertical wavelengths measured by RO may be expected.

## 1. Introduction

The Global Positioning System (GPS) radio occultation (RO) technique has proven to be a powerful tool
to analyze meteorological tropospheric features with a moderate/high spatial and temporal resolution in essentially any meteorological condition. Its ability to penetrate clouds allows to retrieve temperature ($T$) and water vapor pressure ($e$), amongst several other atmospheric variables with high vertical resolution close to the near surface. Vertical profiles of $T$ are provided with an accuracy better than than1 K (e.g., Kursinski et al., 1997; Steiner and Kirchengast, 2005; Scherllin-Pirscher et al., 2011; 2017; Kursinski and
Gebhardt, 2014) in the troposphere to lower stratosphere and specific humidity with an accuracy of about 0.1 g kg$^{-1}$ to 0.3 g kg$^{-1}$ in the lower to middle troposphere. Although measurements are taken irregularly in time and space, they provide global coverage. Data are available from several missions, such as the CHAllenging Minisatellite Payload (CHAMP), Satélite de Aplicaciones Científicas-C (SAC-C), Gravity Recovery and Climate Experiment (GRACE), or the Formosa Satellite mission-3/ Constellation
Observing System for Meteorology, Ionosphere, and Climate mission (hereafter referred to as COSMIC), and have been found to be of high quality and consistency in the troposphere and lower stratosphere (e.g., Steiner et al., 2011; 2013; Angerer et al., 2017, this AMT special issue).

Biondi et al. (2011) recognized double tropopause events using bending angle ($BA$) anomalies derived from GPS RO measurements from different missions. Later, Biondi et al. (2014) found that the GPS RO
technique is useful for understanding the thermal structure of tropical cyclones and possible overshootings into the stratosphere. The complexity of the relationship between deep convection and flow convergence over mountains has been widely studied. Demko et al. (2009) showed that during days with deep convection, the convergence over mountains is weaker than on days when deep convection does not occur.

Over the Alps, considering the kinematic and dynamic features of divergence, flow splitting or mesoscale vortices, it is possible to find regions which initiate or intensify the storms (Bica et al., 2007). Thermally driven flows over the Alps are associated to convergence caused by large-scale topographic heat flows (Langhans et al., 2011). These flows supply moisture from source regions close to the surface, which in turn stimulates the initiation of deep convection (e. g. Barthlott et al., 2006). Gladich et al. (2011)
mentioned that southward oriented reliefs receive more solar radiation, resulting in a warmer atmosphere in comparison with flat terrain. Also, the orography presents a negative energetic balance as compared to flat regions.

Extratropical regions in the Southern Hemisphere show strong wave activity close to the Andes and to the Antarctic Peninsula (e.g., Eckermann and Preusse, 1999; de la Torre et al., 2012; Hierro et al., 2013). The
eastern side of the Andes at mid-latitudes, between the subtropical and polar jets (Houze, 2012), is a natural laboratory for gravity waves (GW), in particular mountain waves (MW). The dynamic processes involved in convection over this region have been analyzed, e.g., by de la Torre et al. (2004), who found

that anabatic winds act as triggering mechanism in the presence of moist enthalpy under unstable conditions. A relationship between MW and the development of deep convection was found by de la Torre et al. (2011). Through the design of several non-dimensional numbers related to storms development and MW energy, Hierro et al. (2013) found that MW are able to provide the necessary energy to overcome a surface stable layer. Vertically propagating short-period GW strongly affect the general circulation as well as the structure of the middle atmosphere (e.g., Dutta et al., 2009).

Convective activity is one of the most important sources of GW through the release of latent heat, contributing to the interaction between waves and mean flow in the middle atmosphere (e.g., Alexander, 1995; Pandya and Alexander, 1999). When a convective cloud reaches the mean flow, waves which propagate upstream are generated (Beres et al., 2002 and references therein). Convective instability, in turn, yields oscillatory movements, which give place to GW that propagate vertically as a harmonic oscillator (e.g., Fritts and Alexander, 2003). Several authors have analyzed the main mechanisms which describe the possible sources of gravity waves generated by convection. In the "obstacle effect", the background finds a barrier provided by the convective flow (Clarke et al., 1986). Fovell et al. (1992) proposed a mechanism where updrafts and downdrafts reach the tropopause, generating high-frequency GW. Röttger (1991) studied penetrating cumulus convection which generates GW by transferring kinetic energy from the troposphere to the lower stratosphere.

Evan et al. (2012) showed that the Weather Research and Forecasting (WRF) mesoscale model (Skamarock et al., 2008) is able to simulate stratospheric GW, when it is run under actual boundary conditions. It was also possible to resolve GW generated by convection in the tropics. Stephan and Alexander (2014), in turn, showed that WRF physics parameterizations are not decisive to obtain good results from GW simulations. From WRF simulations above the Southern Andes, de la Torre et al. (2012) detected systematic large-amplitude, stationary, nonhydrostatic GW structures, forced by the mountains up to the lower stratosphere and persisting for several hours. Their dominant modes were characterized by horizontal wavelengths ($\lambda_H$) of around 50 km. The vertical wavelengths ($\lambda_Z$) were estimated to be between 2 km and 11 km. Over the Andes region, de la Torre et al. (2011) detected two main modes of mountain waves with large amplitude and high intrinsic frequency. Over the same region, Hierro et al. (2013) found stationary modes with $\lambda_H$ between 40 km and 160 km and $\lambda_Z$ of around 7 km. de la Torre et al. (2015), analyzing storms in the presence of MW, distinguished two different structures in vertical wind simulations. Both of them seem to be fixed to the mountains, defining systematic updraft and downdraft sectors. GW parameters were analyzed from band-pass and wavelet analysis, indicating for the cases analyzed the presence of short $\lambda_H$ and long $\lambda_Z$, as expected for high intrinsic frequency GW.

The motivation of the present work is twofold: First, to find a set of collocations among GPS RO *BA* and *T* profiles and mesoscale sub-tropical convective systems under reasonable conditions of proximity in space and time occurrence over orographic regions (Alps and Andes). We use this dataset to test the utility of Global Positioning System (GPS) radio occultation (RO) observations for the detection of convective systems. To detect the cloud top altitude from the RO profiles, we apply a technique based on the anomaly in the *BA*. Secondly, the GW structures are analyzed and discussed for two selected case studies detected in the absence of jets and fronts, from high resolution mesoscale model simulations and

reanalysis data. The possible determination or misleading interpretation of GW parameters from GPS RO is discussed in detail for these case studies. Section 2 outlines the RO data used and the methodology applied and describes the two subsets of RO events retrieved during convective activity close to the Alps

and Andes mountain ranges. In Section 3, one case study at each region is selected and relevant GW features are analyzed from the simulation, the reanalysis data and the measurement of both RO events. In Section 4, conclusions are given.

## 2. Data and methodology

The utility of RO observations for the investigation of convective systems (Biondi et al, 2012; 2015b) and GW over orographic regions in Europe and South America (Alps and Andes mountains) is tested. It is known that a sharp spike in RO *BA* is highly correlated to the top of the cloud, corresponding to anomalously cold temperatures within the cloud. Above the cloud, the temperature returns to background conditions, and a strong inversion appears above the cloud top. For usual tropospheric cloud tops, the *T*

lapse rate within the cloud often approaches a moist adiabat, consistent with rapid undiluted ascent within the convective systems.

We built a collocation database between RO observations and mesoscale convective systems over subtropical to mid-latitude mountain regions. The selected regions for the Alps and Andes are [40°N to 55°N, 0°E to 20°E] and [20°S to 40°S, 60°W to 74°W], respectively (Fig. 1). We use RO data processed at the

Wegener Center for Climate and Global Change (WEGC) with the Occultation Processing System (OPS) version 5.6 (Schwärz et al., 2016), based on excess phase and orbit data version 2010.2640 from the University Corporation for Atmospheric Research (UCAR) from the CHAMP, SAC-C, GRACE, and COSMIC missions. We analyze *BA* and *T* profiles which are available from near surface up to 40 km altitude with 100 m vertical sampling.


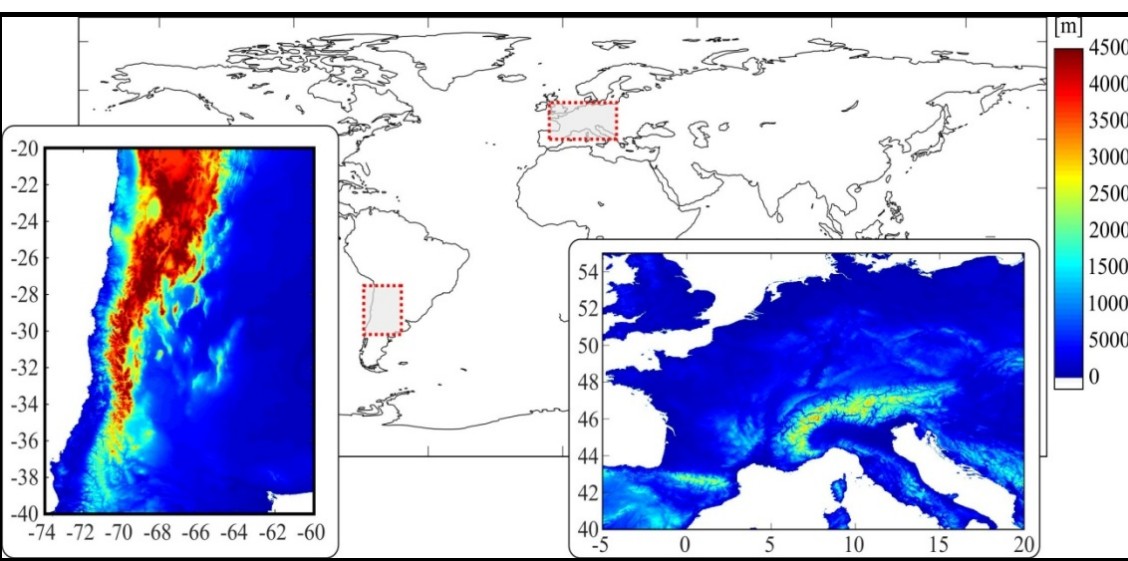

Figure 1: The Alps (right) and Andes (left) regions in Europe and South America, selected to build a collocation database between RO data and cloud data over sub-tropical to mid-latitude mountain regions. The elevation map for each region is included.

Convective systems are located in time and space using the global deep convective tracking database of the International Satellite Cloud Climatology Project (ISCCP) (Rossow et al., 1996), from January 2006 to July 2008. This period was chosen due to the constraints and limitations imposed by the ISCCP database and the COSMIC data. The first source, available between 1983 and 2008, is currently incomplete and being re-processed. The global ISCCP data set, with a horizontal grid resolution of 30 km

and a nominal time resolution of 3 h, is based on brightness temperatures from geostationary satellite measurements. It provides information on the location and extent of mesoscale deep convective cloud systems and their properties. The parameters extracted from ISCCP data, are: time of occurrence, center (mass center) and radius of the storm. The COSMIC mission started in June 2006.

The selection criterion applied in the present work considers the position of the RO observation with

respect to the center of the storm, thus providing 294 and 50 collocations at the Alps and Andes regions respectively. According to this criterion, it is observed whether the latitude and longitude corresponding to the mean tangent point (TP) belonging to each RO profile are located within a radius of 100 km with respect to the center of the storm, as provided by ISCCP. A maximum time difference of 3 h was allowed between each RO event and the data from ISCCP. The collocated events were selected using cloud data

from geostationary satellites METEOSAT (Europe) and GOES (South America) (Fig. 2).

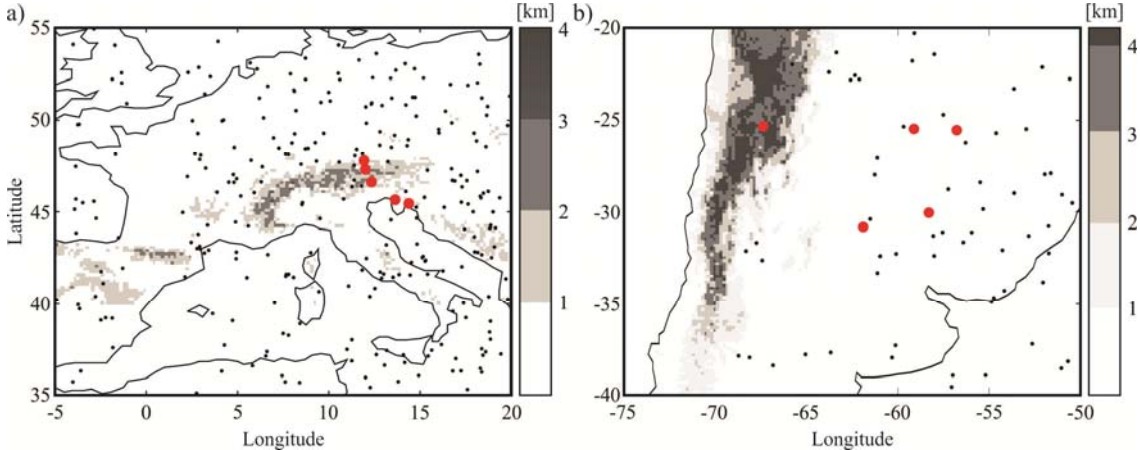

Figure 2. Collocated events between GPS RO profiles and convective systems (a) in the Alps region and (b) the Andes region. The latitude and longitude corresponding to the mean tangent point belonging to each RO are located within a radius of 100 km with respect to the center of the storm and in a time

interval less than 3 hours. Events considered in this study are indicated in red.

The collocated RO $BA$ and $T$ profiles were also used to determine the vertical structures of sub-tropical convective systems over orographic regions. In order to detect the cloud top altitude with RO, we applied the anomaly technique developed by Biondi et al. (2013) atmospheric $BA$ profiles for cloud top detection

of convective systems. Each $BA$ and $T$ profile collocated with a storm was referenced against a RO

background reference climatology profile, which was extracted for the same location and the same month from the global RO *BA* and *T* reference climatology, respectively (for details see Biondi et al. (2017)). We then subtracted the collocated RO reference climatology profile from the individual RO profile. *BA* was normalized with respect to the reference climatology profile in order to obtain a fractional anomaly 170 profile. The cloud top altitude is represented as a pronounced spike in the vertical *BA* anomaly structure and correspondingly, in the *T* anomaly profile.

Taking into account the wave signatures observed in the RO *T* profiles (see below), GW were analyzed from two different data sources: mesoscale model simulations and European Centre for Medium-Range Weather Forecasts (ECMWF) Reanalysis Interim (ERA-Interim) data. The WRF simulations (Skamarock 175 et al., 2008) were performed using 1°×1° National Center of Environmental Prediction (NCEP) Global Final Analysis (FNL) as boundary conditions. They are conducted in four nested domains (27 km, 9 km, 3 km, and 1 km respectively) with 60 vertical levels. A sponge layer was applied in the upper 3 km. The size of the inner domain in the Alps region is about 300 km x 200 km and that of the Andes region about 300 km x 300 km. For each domain, the microphysical schemes used were the following: WRF Single 180 Moment-6 class (WSM6; Hong et al., 2004); Yonsei University (YSU; Hong et al., 2006) to represent the planetary boundary-layer (PBL) physics; Rapid Radiative Transfer Model Longwave (RRTM; Mlawer et al., 1997) and MM5 Dudhia Shortwave (Dudhia scheme; Dudhia, 1989) for radiation processes; the Noah land surface model (developed jointly by NCAR and NCEP; Skamarock et al., 2008) and Monin–Obukhov scheme; (Monin and Obukhov, 1954) for surface physics and thermal diffusion processes, 185 respectively. The cumulus parameterization used was the New Grell scheme (Grell3; Grell and Devenyi, 2002) for the first two domains, while no-cumulus parameterization was selected for the two inner ones.

In the present study, a GW climatology from the limited available number of collocated cases is not intended. Instead, by focusing on selected large amplitude RO *T* profiles, wave features at both mountain subtropical regions are compared. In doing so, five collocations were pre-selected in each region (Fig. 3a-190 b). All of the pre-selected collocations show some spikes in the *BA* profile, from which one is correlated with the cloud top of the corresponding convective structure. Large amplitude oscillations possibly associated with hydrostatic and/or non hydrostatic GW structures are evident. The analysis is further conducted on two peculiar case studies (central panels in Fig. 3a-b), defined by the simultaneous absence of jets and fronts.

To confirm this scenario, we analyzed a possible imbalance in the flow between mass and momentum, able to generate inertia-gravity waves through geostrophic adjustment, as the atmosphere tries to restore the equilibrium (see e.g., Zhang et al., 2000; 2004; Plougonven and Zhang, 2014). In doing so, we considered different available methods. Each of them involves the calculation of a specific parameter, with its advantages and disadvantages (cross-stream component of the Lagrangian Rossby number ($Ro\perp$), 200 Psi vector, generalized omega equation, nonlinear balance equation). Following de la Torre at al. (2006), we analyzed the $Ro\perp$ distribution from reanalysis and from the simulated geopotential and velocity data. When a geostrophic imbalance (e.g. Fritts and Alexander, 2003) coexists with other sources, $Ro\perp$, defined by the ratio of the component of the ageostrophic wind normal to the flow to the observed wind speed, is expected to be greater than 0.5, and the analysis should be more intricate. In the present case studies, $Ro\perp$

remains < 0.5 (not shown) at jet pressure levels, and the GW structures are conceivably limited to orographic and/or convective sources. The RO case studies exhibit attractive oscillatory features and correspond, in the Alps region, to 02 Feb 2008, 17:24 UTC (47.29°N, 12.02°W) and in the Andes region to 19 Dec 2006, 16:56 UTC (25.35°S, 67.37°W).

a)

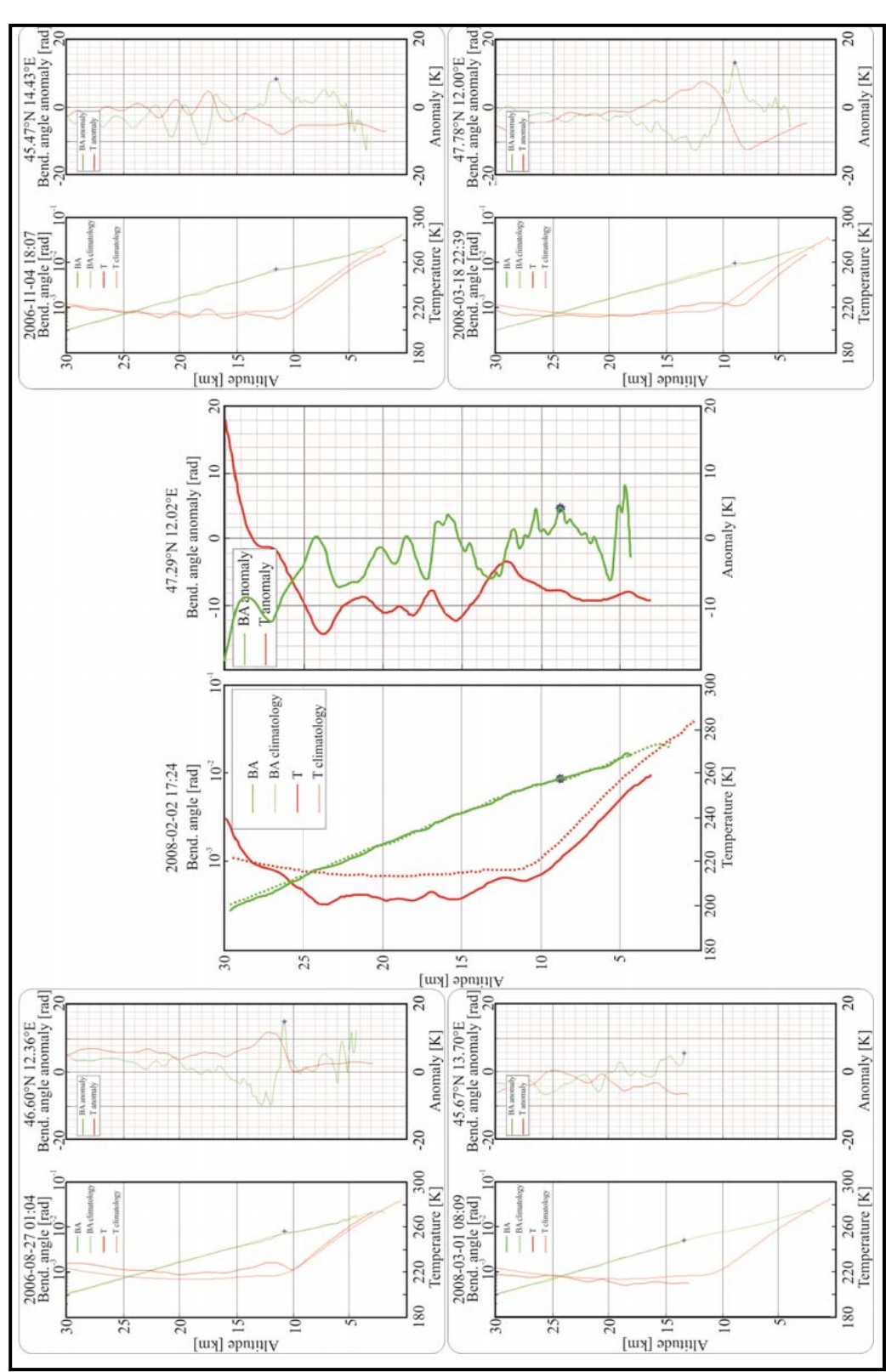

b)

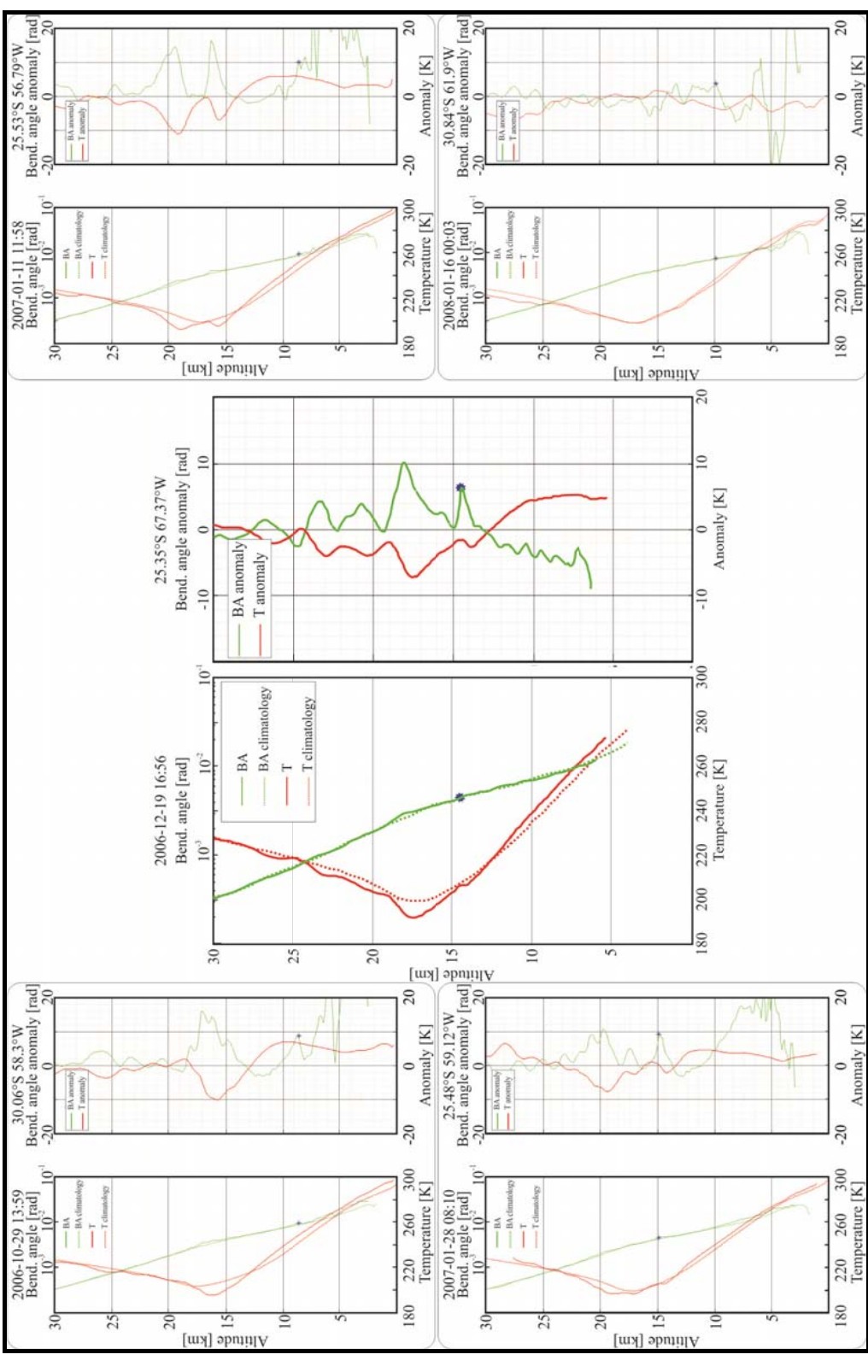

Figure 3: Pre-selected collocations between GPS RO profiles and convective developments in a) the Alps region and b) the Andes region. The figure shows absolute profiles and anomaly profiles for bending angle (green) and temperature (red) and the respective climatology profile (dotted). The cloud top height is indicated by a blue star. The pre-selected collocations for case studies are based on large amplitude wave features in observed the RO $T$ profiles. The two selected case studies (central panels) are based on the simultaneous absence of GW sources given by jets and fronts.

**3. WRF model simulations, ERA Interim reanalyses data and GPS RO observations**

The GW in the two selected study cases at the Alps and Andes region are investigated from WRF numerical simulations, ERA Interim reanalyses data and the collocated GPS RO observed profiles.

3.1. Case study over the Alps region

3.1.1. *Numerical simulations of GW structures*

At the Alps region, the dynamic and thermodynamic parameters are simulated. In Fig. 4, the vertical air velocity ($\delta w$) in the considered area is shown. Two altitude levels (8 km and 12.5 km) are chosen, above and below the cloud tops (situated at 9.8 km height) at 17:00 UTC. The $\delta w$ field is represented a few minutes before the RO event (17:24 UTC).

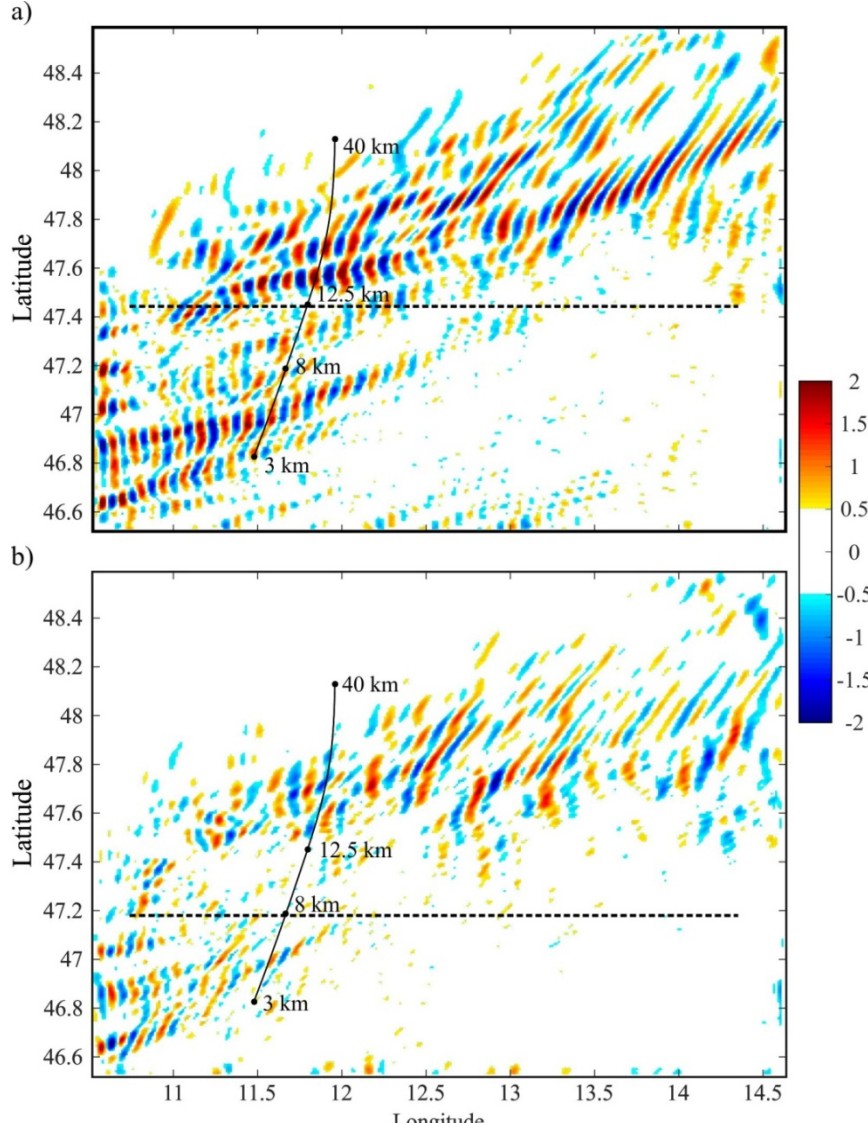

Figure 4: Simulated high resolution (inner WRF domain) $\delta w$ structures at the Alps region (defined in Fig. 1). Two altitude levels, a) above and b) below the clouds top, are shown. The line of tangent points (LTP) corresponding to the collocated RO event is indicated in both panels from lower (3 km) to upper (40 km) points (solid black line). The line of sight (LOS, dotted black line) at both levels is also indicated.

The mesoscale outputs were obtained every 60 min. It is generally accepted that $\delta w$ adequately highlights the main GW amplitudes and wavelengths characteristic of MW. These usually belong to high and medium intrinsic frequency regimes. In Fig. 4, coherent, mostly bi-dimensional GW structures with constant phase surfaces are seen. They are mainly oriented along S-N direction and slightly tilted to N-E with increasing latitude. The mean horizontal wind is directed from NW to SE at 700 hPa, causing the apparent forcing of MW. It is equal to $[U,V] = [3;-3]$ m/s at 18 UTC. Prevailing amplitudes and $\lambda_H$, ranging between 1-2 m s$^{-1}$ and 20-60 km respectively, are distinguished. Two main features may be remarked at both levels. GW amplitudes are weaker below than above the cloud tops and two different structures are visible. One structure is stationary and the other is not stationary. The last one is zonally

and meridionally displaced when observed at 1-hour intervals during the evolution of convection (not shown). The GW sources seem to be orographic forcing and associated to cloud development. $\delta w$ amplitude values up to 2 m s$^{-1}$ correspond to MW with short horizontal wavelength. These $\delta w$ perturbations exhibit, as expected, the presence of MW more clearly than $\delta T$. (Fig 5a). By contrast, the non stationary GW with longer $\lambda_H$ and amplitude values above 2 K are more evident in $\delta T$ than in $\delta w$, as a function of longitude and latitude (Fig 5b and 5c). Systematic $\lambda_Z$ values ~ 8 km, associated to these longer $\lambda_H$ are observed. In these figures, the vertical line indicates the position of the RO mean TP. The orographic amplitudes are more significant early in the morning, exhibiting a general decrease with increasing local time (not shown). They reach large amplitudes at stratospheric heights beyond the tropopause, located at 11 km. No critical levels for MW or reflection effects are observed (Fig.5d). The strongest orographic structures are observed until the early afternoon. On the other hand, the non-stationary GW packets are generated between 12 km and 17 km height, during the convection development after mid-afternoon, and radiated above the cloud tops. The longer $\lambda_H$ values are not well defined, suggesting the coexistence of two or more non-stationary modes.

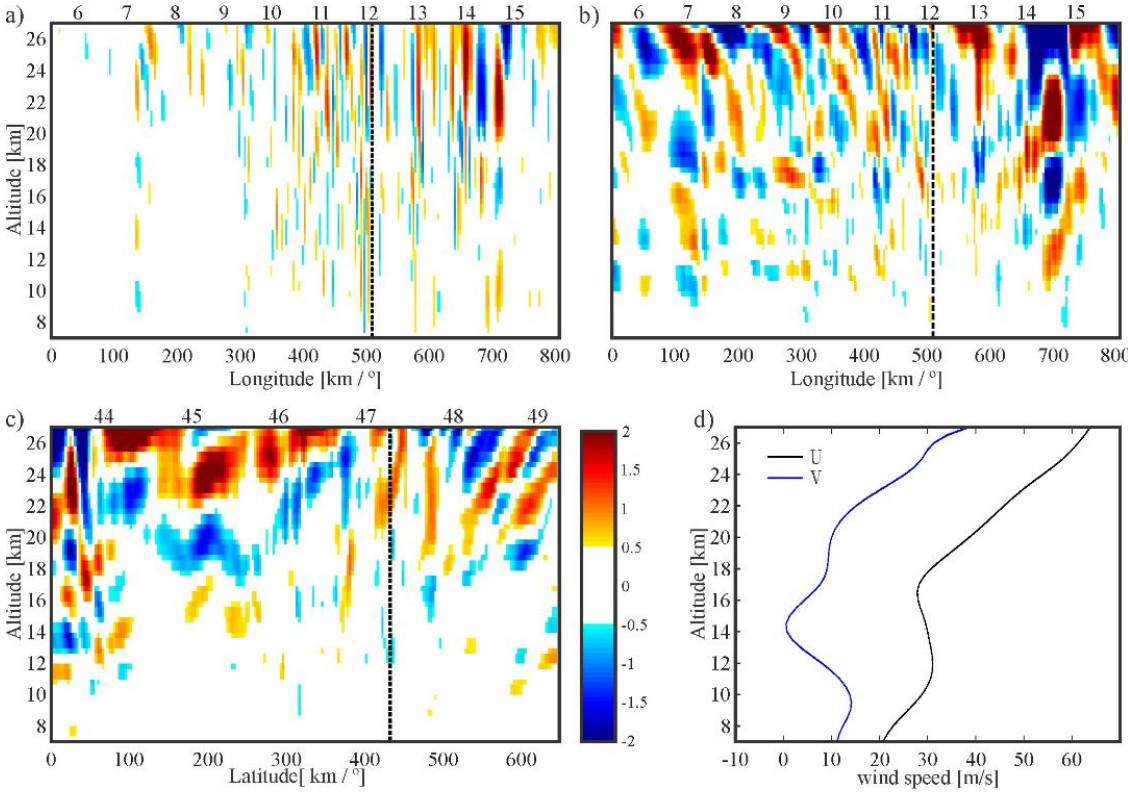

Figure 5. Simulated high resolution GW structures at the Alps region, showing a) $\delta w$ and b)-c) $\delta T$ signatures as a function of longitude and latitude, respectively. Non-stationary/stationary GW with longer/shorter horizontal wavelength are more clearly seen in $\delta T/ \delta w$. The dotted line indicates the position of the TP. d) Zonal and meridional mean wind.

*3.1.2. Analysis of gravity waves in the RO observation*

The wavelike structure of the RO *T* profile retrieved at the Alps region is analyzed. This profile is shown in the central panels of Fig 3a. Its horizontally projected line of tangent points (LTP) is seen in Fig. 4.

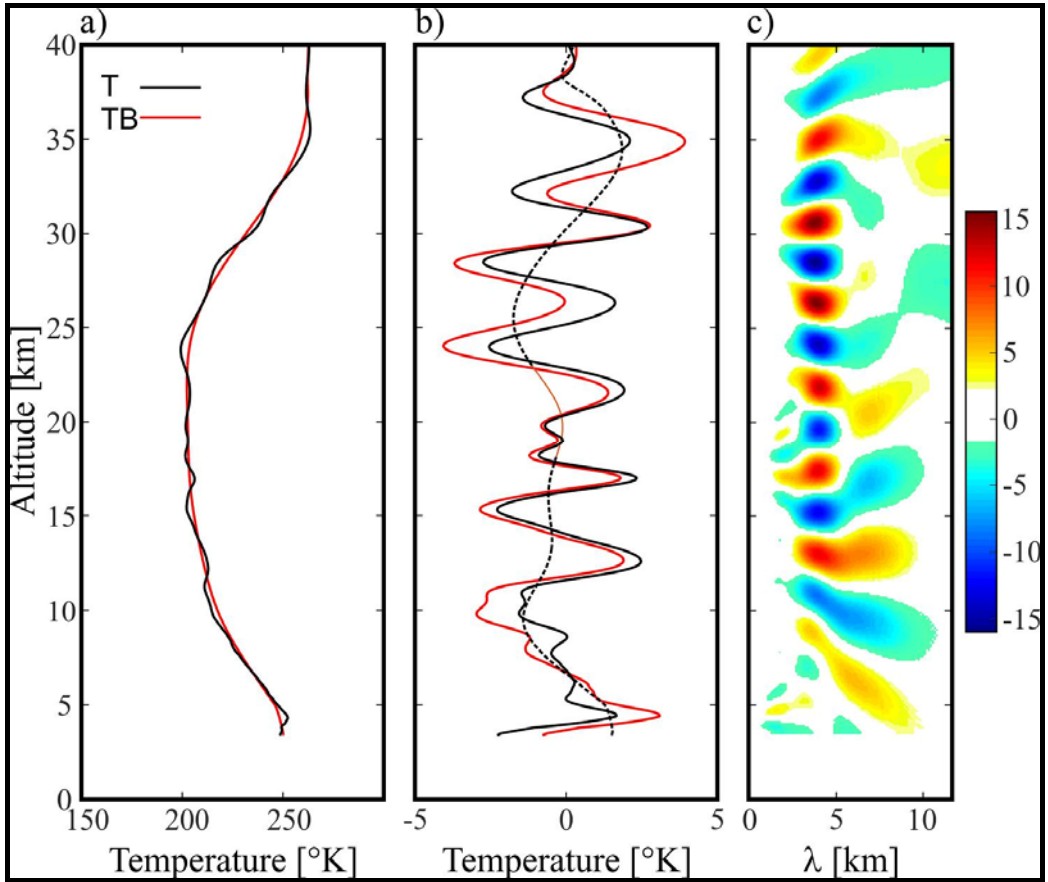

Figure 6: a) Retrieved RO *T* vertical profile (black line) and its corresponding low-pass background
component *TB* (red line) at the Alps region. b) Remaining oscillation (black), isolated after applying a band-pass filter (red) and a double filtering process to the background (black dotted) profile. c) Continuous wavelet transform (CWT) applied to the perturbation $\delta T$ structure, as a function of the vertical wavelength. The RO event took place at 17:24 UTC. The values at the right of the color bar represent the degree of correlation between the wavelet selected (Morlet) and the signal to be analyzed
($\delta T$).

In doing so, the perturbation component of this profile is removed (Fig. 6a). If a simple band-pass filter is applied (e.g., de la Torre et al, 2006), the "tropopause problem" must be dealt with, as far as RO *T* profiles are available as a function of altitude. As pointed out by Alexander et al. (2011), even if filters
with ideal cut-offs existed, part of the problem would still be there because the tropopause kink usually departs from a sinusoid or any other function that may be used as a basis. Real filters do not yield ideal spectral component isolations (one side effect is amplitude reduction, for example) and may need some

manual fine tuning procedures to optimize their performance. After a "perfect" band-pass is used (able to completely filter out the undesired wave modes), there should be no remaining components at wavelengths outside the considered range. The method applied in this case comprises two steps: (i) a bandpass filtering is used to isolate the wavelength range of interest in order to separate the background and to eliminate the noise. Then (ii) a cutoff larger than or equal to the bandpass upper limit is applied. This allows to remove large wavelengths representing background behavior or trends still present and to force a zero mean. The tropopause kink in $T$ can be viewed as the surrounding of a long sinusoidal peak. In the first step, a bandpass between 1 km and 10 km, and in the second step, a cutoff of 10 km are applied. The resulting filtered $\delta T$ profile is marked by the solid black line in Fig. 6b. A continuous wavelet transform (CWT) is finally applied to the remaining $\delta T$ wave structure (Fig. 6c).

The CWT is a useful method to detect the main oscillation modes present in a signal analysis. As is known, it is a powerful tool for studying multiscale and non-stationary processes occurring over finite spatial and temporal domains (Lau and Weng, 1995). It allows detecting short-period as well as long-period oscillations. The CWT compares the original signal against a set of synthetic signals, called mother wavelets, obtaining correlation coefficients. The comparison between signals is carried out through a process of translation and contraction or dilation of the mother wavelet in each signal portion. This process is repeated for all scales of mother wavelets, allowing the location of short life, high-frequency signals like sharp changes, thus obtaining detailed information. In this work, the mother wavelet selected is the Morlet wavelet (Morlet, 1983), which consists of a flat wave modified by a Gaussian envelope. Fig. 6c shows a clear GW signal throughout the tropo-stratospheric region, with prevailing $\lambda_Z = 4$ km. A second considerably weaker mode is also present in the troposphere, with $\lambda_Z$ close to 7 km.

To search for a possible correspondence among these two modes and the structures observed in Figs. 4 and 5, the expected amplitude attenuation effects must be considered (Alexander et al., 2008). For this RO event in particular, the LOS stands at each TP almost exactly perpendicular to the GW phase surfaces observed in Fig. 3, at 88° from north direction (dotted lines in Fig. 4). This should prevent the observation of vertical oscillations corresponding to short $\lambda_H$ structures, as seen in Fig. 4. In this figure, the quasi perpendicular orientation of the constant GW phases relative to the line of sight (LOS) is observed. This clearly does not benefit the GW detection during the RO event. The horizontal averaging of RO retrievals produces an amplitude attenuation and phase shift in any plane GW, which may lead to significant discrepancies with respect to the original values (Alexander et al., 2008). The lower and upper altitudes of LTP are 3 km and 40 km, respectively. The observation of GW structures with short $\lambda_H$ observed in Fig. 4 is expected to suffer amplitude attenuation and should not be visible during GPS RO events (Preusse et al., 2002; Alexander et al., 2008).

An estimation of the expected attenuation in the stationary and non-stationary structures during the RO sounding is thus performed. The amplitude attenuation factor defined as the ratio between derived and original amplitudes is deduced. This factor is a function of the ratio of GW vertical and horizontal wavelengths and the angle (on the horizontal plane) between the wave fronts and the line of sight (LOS). Ideal conditions that lead to no attenuation are, respectively, a null ratio between vertical and horizontal wavelengths and a null angle between LOS and the fronts. For the analyzed case study, Table 1 shows the

corresponding values of both parameters and the attenuation factor (range 0 to1 covers from null to full output). According to these results, in the Alps case study a considerable attenuation of mountain waves below the tropopause and above it, is expected.


### 3.1.3. Analysis of GW structures from ERA Interim data.

The resulting estimation of a negligible attenuation factor at both height levels shown in Fig. 4 confirms that the GW cannot be captured there during the RO event. It is clear that the mesoscale simulations are not enough to explain the observed GW structure. To understand the origin of the oscillation observed in

the RO $\delta T$ profile, the possibility is analyzed that the presence of large scale GWs cannot be captured by mesoscale WRF simulations. According to this, the corresponding ERA Interim reanalyses data are analyzed. In Fig. 7, $\delta T$ resulting from these data at 22 km height reveals a well defined non-stationary oscillation propagating in NW-SE direction, with $\lambda_X$ and $\lambda_Y$ equal to 530 and 650 km respectively. ERA Interim provides information about these horizontal wavelength GW, but has a relatively coarse

resolution, strongly underestimating wave amplitudes. Accordingly, amplitudes in Fig. 7 are quite low. If this wave is seen in RO soundings, it would have a much larger amplitude. The attenuation factor derived from these values is estimated to 0.90 (first line in Table 1). This value is consistent with a clear oscillation observed at stratospheric heights in the RO profile, with $\lambda_Z$ equal to 4 km (Fig. 6c and Table 1).


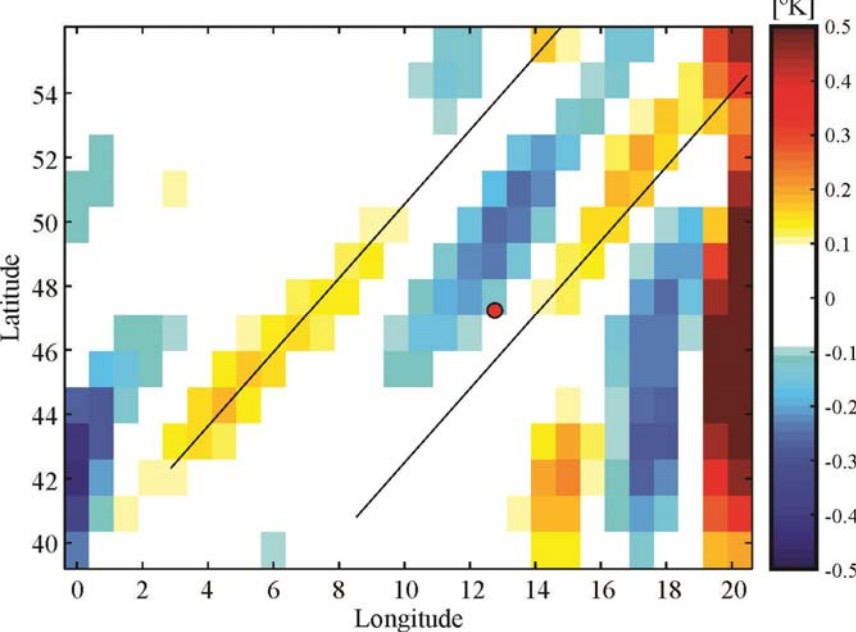

Figure 7. ERA Interim $\delta T$ data corresponding to the case study at the Alps region. The red dot indicates the mean TP location.

|  | Vertical/horizontal wavelength ratio | Angle between wave fronts and LOS | Attenuation factor |
|---|---|---|---|
| Alps above TP | 0.02 | 48 | 0.90 |
| Alps below TP | 0.51 | 78 | 1e-49 |
| Andes above TP | 0.01 | 32 | 0.99 |
| Andes belowTP | 0.35 | 7 | 0.44 |


Table 1. Attenuation factor as a function of the ratio of GW vertical and horizontal wavelengths and the angle (on the horizontal plane) between the wave fronts and the LOS for the Alps and Andes case studies above and below the tropopause level (TP). A considerable/partial attenuation below the tropopause at the Alps/Andes region is expected. Line 1 in the table refers to the wave seen in ERA-Interim (Fig. 7), while

lines 2 to 4 in this table refer to the WRF simulations.

Regardless of the expected attenuation, an additional distortion must be considered due to the slanted nature of the sounding. This is introduced in the measured GW $\lambda_H$ and $\lambda_Z$ by atmospheric soundings performed in any direction other than vertical and horizontal. This is the case during GPS RO events or during radiosoundings (e.g., de la Torre and Alexander, 1995; Alexander et al., 2008). In the case of RO

events, a visibility condition imposed to the line of sight (LOS) described in P. Alexander et al. (2008), must be satisfied. The distortion is more or less significant, depending on the elevation angle of the sounding path and the GW aspect ratio (de la Torre et al., 2018, hereafter referred to as T18). For example, during vertically directed measurements with a single lidar, $\lambda_H$ cannot be detected but $\lambda_Z$ is not distorted. A clear advantage of numerical simulations or reanalyses data is that they are not affected by

this systematic error inherent to any slanted atmospheric measurement. According to this, we must distinguish between "apparent" (i.e., RO observations) and "real" $\lambda_H$ and $\lambda_Z$ values. For this reason, the apparent 4 km oscillation observed in Fig. 6c must be carefully observed as a distorted signature, which in fact corresponds to a different real $\lambda_Z$ value (T18). Considerable under- or overestimations are generally expected, depending on the aspect ratio of GW and inclination of the LTP (T18 and Appendix).


3.2. Case study over the Andes region

3.2.1. *Numerical simulations of GW structures*

From the simulated dynamic and thermodynamic parameters, we show $\delta w$ at constant height levels for the Andes area in Fig. 8. Constant altitudes of 10 kmand 16 km, below (Fig. 8a) and above (Fig. 8b) the

cloud top (situated at 14 km height), respectively, and at 26 km (Fig. 8c), at 17:00 UTC, are selected. The $\delta w$ field is represented a few minutes before the RO event (16:56 UTC).

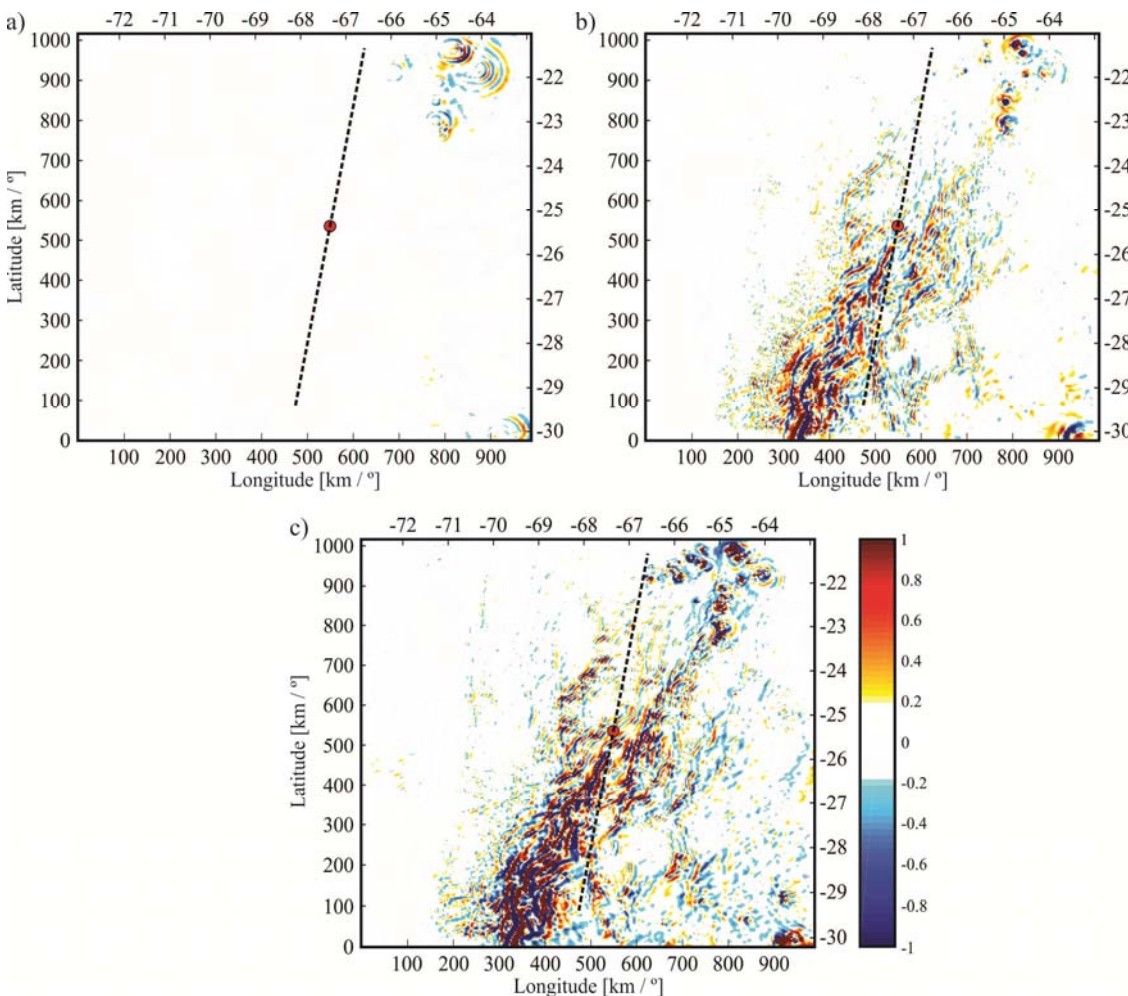

Figure 8: Simulated high resolution GW structures in the Andes region defined in Fig. 1. Three constant altitudes, a) at 26 km, b) at 16 km (above) and c) at 10 km (below) the cloud tops (situated at 14 km) are
selected. In this case, the mean LTP (black line) is located around 25.5°S and 67.1°W. The red point indicates the mean TP. The dotted line indicates the LOS crossing the TP corresponding to each selected altitude.

Coherent bi-dimensional GW structures with constant phase surfaces oriented from SW to NE are seen
(Fig 8a and 8b). The mean horizontal wind vector at 700 hPa directed from NW to SE is [U,V] = [7;-3] m/s at 18 UTC. de la Torre et al. (2015) have shown that immediately to the south of Central Andes, close to the mountain tops, two clearly different orographic GW structures are systematically observed at constant pressure levels. One structure type shows highly elongated alternating positive and negative parallel $\delta w$ bands, aligned almost in N-S direction. The second type presents a bi-dimensional distribution
too, but exhibits alternating fringes of much shorter longitude, mostly in SW–NE direction. The mean wind that forces mountain waves exhibiting the 1D structures of the first type presents an intense zonal gradient of zonal wind, veering to an increasing westerly mean wind with increasing latitude. The

meridional wind component is usually negligible. In the 2D structures of the second type, at and below mountain top levels, a prevailing intense negative meridional wind with less zonal wind contribution is observed. In the case study shown in Fig. 8, the mean wind at 700 hPa with considerable negative meridional component is associated to the second type. In Fig. 8a to 8c, additional non-orographic GW structures situated at the NE and SE of the domain are observed. They propagate from low altitudes up to at least the upper limit of the simulations. These structures, probably of convective origin, exhibit stationary circularly shaped wave fronts. They penetrate beyond the critical layer for orographic GW, situated at an almost zero wind level, near to 18 km height (Fig. 8a to 8c and 9d).

In Fig. 9a to 9c, as in the Alps case, $\delta w$ perturbations (Fig 9a) exhibit the presence of mountain waves more clearly than $\delta T$. Nevertheless, in Fig 9b and 9c, the presence of two different GW structures, separated by a critical layer at 18 km as a function of longitude and latitude, is identified. The MW propagate upwards and just below the critical layer they increase their amplitude, with decreasing $\lambda_Z$ in agreement with linear GW theory. Inversely to the Alps case study, the orographic amplitudes at the Andes region are more significant in the afternoon. They exhibit a general increase from sunrise with increasing local time. Above the critical layer, non-stationary GW packets are generated during the convection development after mid-afternoon and are radiated above the cloud tops. These longer $\lambda_H$ values are well resolved by WRF and shorter from the simulations, than in the Alps case study. In the Andes case, there is almost no spatial coexistence between GW from both GW sources. A clear separation between orographic structures below the critical layer and the convective GW above is quite evident.

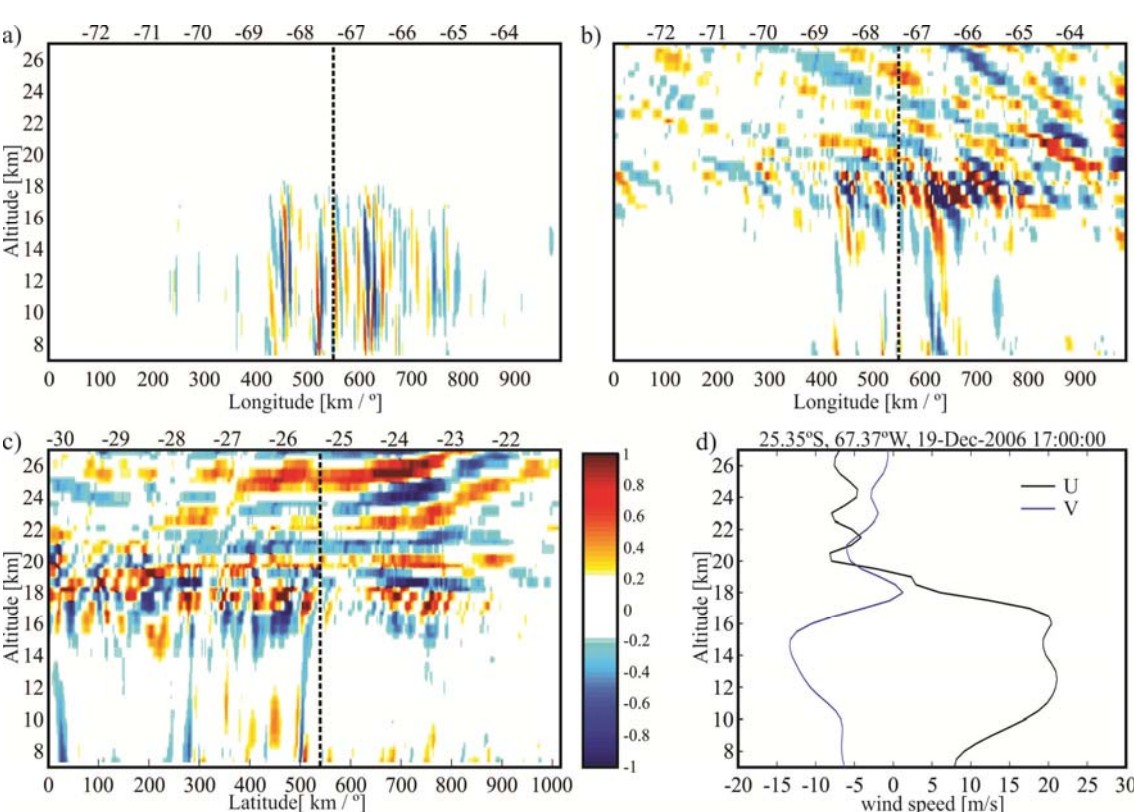

Figure 9. Simulated high resolution GW structures at the Andes region, showing a) $\delta w$ and b)-c) $\delta T$ signatures as a function of longitude and latitude, respectively. Non-stationary/stationary GW with longer/shorter horizontal wavelength and shorter/longer vertical wavelength, above/below the critical layer are observed. d) Zonal and meridional mean wind.

Next, the wavelike structure of the RO $T$ profile retrieved at the Andes region is analyzed. This profile is shown in the central panels of Fig 3b, and its horizontally projected LTP is seen in Fig. 8.

### 3.2.2. *Analysis of gravity waves in the RO observation*

The procedure is identical to the one applied in case study 1. The utility of the double filter applied is
more obvious here than in the Alps case study. Also, the tropopause kink is sharper (Fig. 10b). For this RO event, inversely to the situation described in the Alps case study, the LOS stands at each TP almost aligned to the GW phase surfaces observed in Fig. 8, at 190° from north direction (dotted lines in Fig. 8). This allows us to detect vertical oscillations in the RO profile corresponding to the short $\lambda_H$ MW structures seen below the critical layer in Fig. 8 and 9.

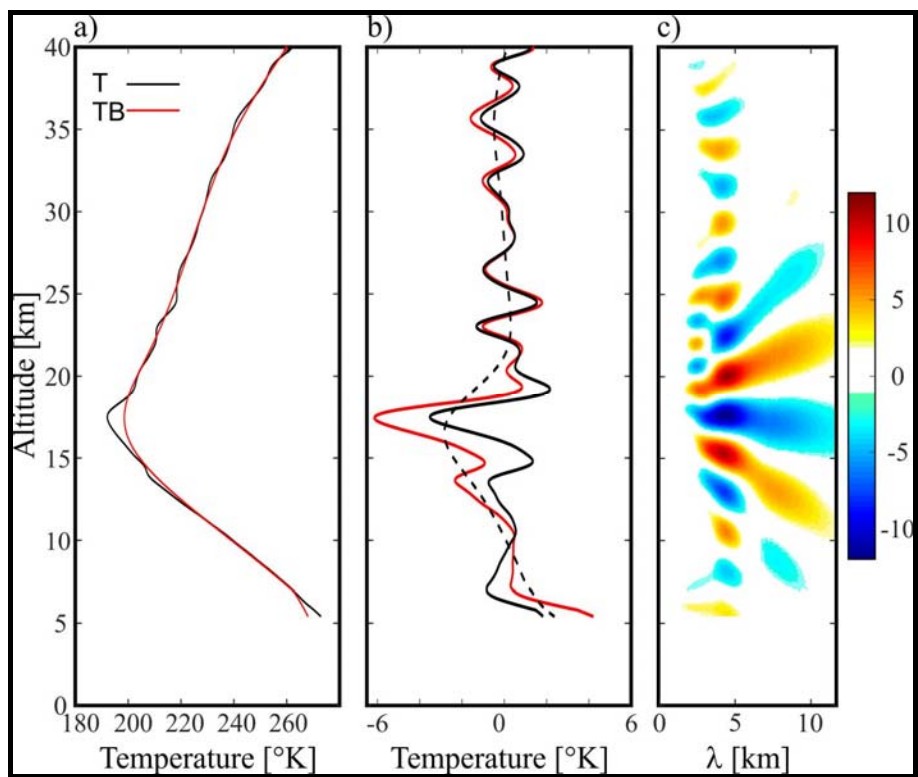


Figure 10: Similar to Fig. 6, but for the RO case study over the Andes region.

In Fig. 10c, a clear GW signal appears also in the Andes case, propagating throughout the tropo-stratospheric region with a prevailing $\lambda_Z = 4.5$ km. Here again, attenuation and distortion effects must be
considered. The computation of a partial attenuation factor (0.44, Table 1) below the tropopause confirms

that the mountain waves at these levels can be captured during the RO event. In this estimation, $\lambda_X$, $\lambda_Y$ and $\lambda_Z$ values equal to 30, 100 and 10 km, respectively, are considered. In this case study, the mesoscale simulations are sufficient to explain the observed GW structure above the critical layer, too. Clear signatures of $\lambda_X$, $\lambda_Y$ and $\lambda_Z$ equal to 350, 400 and 3 km, respectively, may be seen. These values yield a

very high attenuation factor (0.99). Note that attenuation factors close to 1 have to be taken with some caution because "ideal" wave patterns are assumed for this calculation. To understand the origin of the oscillation observed in the RO $\delta T$ profile, we conclude that below the tropopause, the oscillation is completely due to orographic waves and above, due to convective GW. In both cases, mesoscale simulations were able to capture the GW structures. In this case study, like in the Alps case, an expected

distortion in the measured $\lambda_Z$ value, due to the slanted nature of the sounding, must also be expected.

## 4. Summary and conclusions

From an initial set of collocations between convective systems and GPS RO observations, the applicability of these data sets for the detection and investigation of convective systems and GW over

orographic regions was analyzed. In doing so, mountain regions of Europe and South America, over sub-tropical to mid-latitude regions, the Alps and Andes mountain ranges, were selected. A collocation database was built up. We used RO bending angle and $T$ profiles retrieved at the Wegener Center with processing version OPSv5.6. The storm systems were located in time and space according to the global deep convective tracking database ISCCP. From an initial set of 294 and 50 collocations at Alps and

Andes regions, respectively, a subset of 10 collocations was pre-selected. This pre-selection was based on the observation of large amplitudes, presumably GW, in the retrieved $T$ RO profiles. Two case studies were finally studied in detail, at the Alps and Andes regions, respectively. The case studies were investigated using mesoscale WRF simulations, ERA Interim reanalyses data and the measured RO $T$ profiles. The case studies considered were selected based on the absence of jets and fronts, in order to be

able to filter out two relevant possible sources of GW. Similar GW regimes and dominant vertical and horizontal wavelengths, from convective and orographic origin, were found at both regions. MW reach higher altitudes above the Alps than close to the Andes. The background mean wind above the latter region imposes a critical level for mountain wave propagation, preventing the propagation of GW packets up to stratospheric heights.

At the Alps, mostly bi-dimensional GW structures with constant phase surfaces are seen. The mean horizontal wind causes the apparent forcing of MW. Prevailing amplitudes and $\lambda_H$ ranging between 1-2 m s$^{-1}$ and 20-60 km, respectively, are distinguished. GW amplitudes are weaker below than above the cloud tops and two different structures are visible. One structure is stationary and the other is not stationary. The GW sources seem to be orographically forced or associated to cloud development. $\delta w$ amplitude values

up to 2 m s$^{-1}$ correspond to MW with short horizontal wavelength. These $\delta w$ perturbations exhibit, as expected, the presence of MW more clearly than $\delta T$. By contrast, the non-stationary GW with longer $\lambda_H$ and amplitude values above 2 K are more evident in $\delta T$ than in $\delta w$. Systematic $\lambda_Z$ values close to 8 km, associated to these longer $\lambda_H$, are observed. The orographic amplitudes are more significant early in the

morning, exhibiting a general decrease with increasing local time. They reach large amplitudes at stratospheric heights beyond the tropopause, located at 11 km in the Alps case. No critical levels or reflection effects are observed. The strongest orographic structures are observed until the early afternoon. The non-stationary GW packets are generated during the convection development after mid-afternoon and are radiated above the cloud tops. The longer $\lambda_H$ values are not well defined, suggesting the coexistence of two or more non-stationary modes.

The observed RO $T$ profile was first filtered and then a CWT was applied to the remaining $\delta T$. A clear GW signal throughout the tropo-stratospheric region, with prevailing $\lambda_Z = 4$ km, was found. The correspondence between these two modes and the structures observed in the simulations were investigated, considering the expected amplitude attenuation effects in the RO sounding.

It is concluded that the LOS and the wavelength ratio should prevent the observation of vertical oscillations corresponding to short $\lambda_H$ structures. This clearly does not benefit GW detection during the RO event. The horizontal averaging of RO retrievals produces an amplitude attenuation and phase shift in any plane GW, which may lead to significant discrepancies with respect to the original values. An estimation of the expected attenuation in the stationary and non-stationary structures during the RO sounding was performed. The amplitude attenuation factor defined as the ratio between derived and original amplitudes was deduced. Ideal conditions that lead to no attenuation are, respectively, a null ratio between vertical and horizontal wavelengths and a null angle between LOS and the fronts. The case study analyzed shows the corresponding values of both parameters and the attenuation factor. In this Alps case study, a considerable attenuation of mountain waves below and above the tropopause is expected. The resulting estimation of a negligible attenuation factor confirmed that these GW cannot be captured during the RO event. As the mesoscale simulations are not enough to capture and explain the observed GW structure, corresponding ERA Interim reanalyses data were investigated. From these data, a defined non-stationary oscillation propagating in NW-SE direction was observed explaining the oscillation seen in the RO $\delta T$ profile.

At the Andes region, coherent bi-dimensional GW structures with constant phase surfaces oriented from SW to NE, of a type already reported in a previous work, are seen. Additional non-orographic GW structures situated at the NE and SE of the domain are observed. They propagate from lower altitudes until at least the upper limit of the simulations. These structures exhibit stationary circular wave fronts and penetrate beyond the critical layer for orographic GW, situated at an almost zero wind level, near the tropopause at about 18 km height. As in the Alps case, $w$ perturbations exhibit the presence of MW more clearly than $T$. The presence of two different GW structures, separated by a critical layer at 18 km, is identified. The MW propagate upwards and just below the critical layer their amplitude increases and at the same time $\lambda_Z$ decreases. Inversely to the Alps case study, the orographic amplitudes at the Andes region are, as usual, more significant in the afternoon, exhibiting a general increase with increasing local time. Above the critical layer, non-stationary GW packets are generated during the convection development after mid-afternoon and are radiated above the cloud tops. In this case, there is almost no spatial coexistence between GW from both GW sources.

In the retrieved RO $\delta T$ profile a signal appears also propagating throughout the tropo-stratospheric region with a prevailing $\lambda_Z = 4.5$ km. An evaluation of the partial attenuation coefficient reveals that in the troposphere, MW can be captured during the RO event. In this estimation, $\lambda_X$, $\lambda_Y$ and $\lambda_Z$ values equal to
30, 100 and 10 km, respectively, are considered. For the Andes case study, the mesoscale simulations explain the observed GW structure above the critical layer also. Clear signatures of $\lambda_X$, $\lambda_Y$ and $\lambda_Z$ equal to 350, 400 and 3 km, respectively, may be seen. These values yield a very high attenuation factor (0.99). It is concluded that below the tropopause, the oscillation is entirely due to orographic waves and above, to convective GW. In both cases, mesoscale simulations were able to capture the GW structures. In this case
study, as in the Alps case, additionally, an expected distortion in the measured $\lambda_Z$ value must be foreseen. This is due to the slanted nature of the sounding and also depends on the GW aspect ratio.

**Appendix**

The discrepancy between measured and simulated $\lambda_Z$ (an analogous discussion could also be given
regarding $\lambda_H$, –see T18-) may be quantitatively explained as follows. It may be assumed that RO soundings yield $T$ profiles almost instantaneously in such a way that GWs are "frozen" in space during the entire LTP retrieval. The vertical "real" and "apparent" (or measured) wavelengths ($\lambda_z$ and $\lambda_z^{ap}$, respectively) are related according to the following expression (T18):

$$\lambda_z^{ap} = \frac{\lambda_Z}{abs(1 + \cot(\alpha)\cot(\psi))} \quad \text{(A1)}$$

where $\alpha$ is the elevation angle, defined by a straight sounding path direction and the horizontal plane, and $\cot(\psi)$ is the ratio between the horizontal wavenumber vector ($k_H$) projected on the vertical plane containing the LTP, and the vertical wavenumber $k_Z$. The ratio $k_H/k_Z = \lambda_Z/\lambda_H$ is also known as the GW aspect ratio = tg ($\psi$). We define the distortion as the ratio:

$$D = \frac{\lambda_z^{ap}}{\lambda_Z} \quad \text{(A2)}$$

and plot this parameter, following Eq. (A1), as a function of $\alpha$ and $\psi$ (red line in Fig. A1). For the LTP shown in Fig. 8, considering that both the horizontal and vertical excursion of LTP, as well as the upper and lower altitudes (40 and 3 km), are known, we infer an average $\alpha = 0.68$ rad. A different curve is obviously expected for different $\alpha$ values, although fitting a similar shape. The divergence at high $D$ values is only suggested by plotting its variability up to $D = 5$. The left green circle qualitatively indicates
the $D$-$\psi$ GPS RO sector encompassing the functional relation among the 3 parameters. The right green circle is included, because an uncertainty between $\alpha$ and $\pi-\alpha$ for our estimated aspect ratio still remains from our previous estimation. The white, light gray and gray backgrounds indicate, for reference purposes, the non-hydrostatic, hydrostatic non-rotating and hydrostatic rotating GW regimes, respectively. Both quadrants are separated by a vertical dashed curve. The horizontal dashed line
represents the non-distortion limit, where $\lambda_Z$ and $\lambda_Z^{ap}$ should coincide. According to both possibilities (the internal regions defined by any of the green circles), in the case considered (Fig. 10c), there are two

possibilities, respectively indicating an under- and an overestimation of $\lambda_Z$. This uncertainty can be removed by inspection of Fig. 10 (T18).

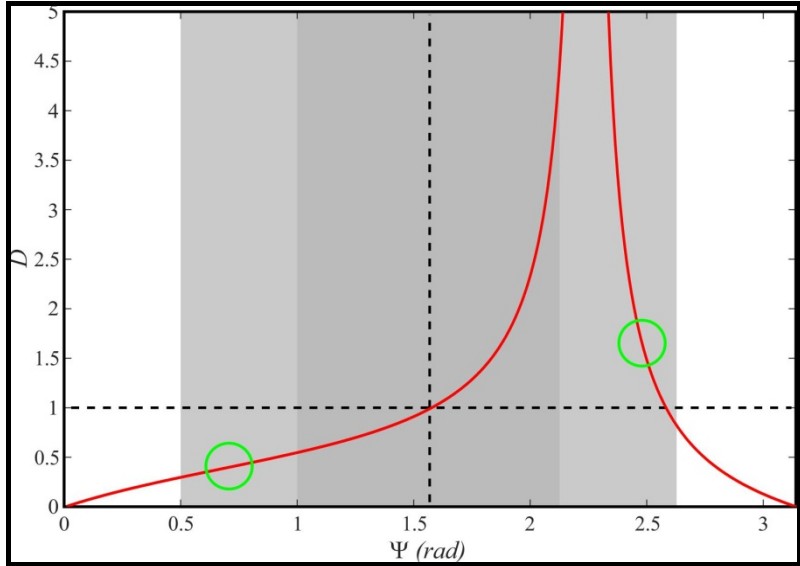


Figure A1: Example of distortion expected between measured and simulated $\lambda_Z$, in relation to the GW parameters derived from the GPS RO event above the Andes, as detailed in Figs. 8, 9 and 10. Both green circles illustrate, for a given $\alpha$ value, possible $D$-$\psi$ combinations of the parameters that would correspond to under- (left green circle) and overestimation (right circle) of $\lambda_Z$ respectively.


**Data availability**

The WEGC OPSv5.6 RO dataset is available on request from WEGC and will be made publicly available in 2018.. Data on convective systems used in this study are available from the global deep convective tracking database of the International Satellite Cloud Climatology Project (ISCCP) via https://isccp.giss.nasa.gov/CT/. Cloud data from METEOSAT are available from the EUMETSAT processing centre via https://eoportal.eumetsat.int/userMgmt/login.faces and data from GOES are available from NOAA via https://www.class.ncdc.noaa.gov/saa/products/welcome. ERA-Interim data are publicly available from ECMWF (Reading, UK) and can be accessed via https://www.ecmwf.int/.

**Author contributions.**

R. Hierro, A. de la Torre, A.K. Steiner, P. Alexander and P. Llamedo designed the study, performed computational implementation and analysis, performed the numerical modeling, created the figures, and wrote the first draft of the paper.

A.K. Steiner provided guidance on RO data and analysis aspects and contributed to finalizing the manuscript.

P. Cremades provided recommendations and assistance in the WRF simulations.

**Competing interests.** The authors declare that they have no conflicts of interest.

**Acknowledgements.** We are grateful to R. Biondi (ISAC-CNR, Rome, Italy) for advice on cloud data and provision of the cloud detection algorithm and the reference climatology. We thank H. Truhetz (WEGC, Graz, AT) for help on WRF model aspects. We acknowledge UCAR/CDAAC (Boulder, CO, USA) for the provision of level 1a RO data and ECMWF (Reading, UK) for access to its analysis, and short-term forecast data. We thank the WEGC processing team members, especially M. Schwärz (WEGC, AT), for OPSv5.6 RO data and his special support. This study was initiated by the Programme of Exchange and Cooperation for International Studies between Europe and South America (PRECIOSA) by funding a research visit of R. Hierro at WEGC (University of Graz, Graz, AT). This paper is funded by the Austrian Science Fund (FWF) under research grant P27724-NBL (VERTICLIM). The study has been supported by the CONICET under grants CONICET PIP11220120100034 and ANPCYT PICT 2013-1097.

**paper is funded by the**

**Austrian Science Fund (FWF) under research grant P27724-NBL**

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
