# Peer review of "Orographic and convective gravity waves above the Alps and Andes mountains during GPS radio occultation events – a case study"

_Atmospheric Measurement Techniques, 2017_

## Referee Comment (RC1) · Anonymous Referee #1 · 14 Nov 2017

**General Comment**

In my AMTD access review several major concerns were raised which showed that a major rewriting and refocusing of the manuscript would be required, probably exceeding the time usually allotted for revisions of a paper under review. Therefore my recommendation at this stage was to reject the paper. Since then, only minor changes were made. This is why this recommendation is repeated.

Of course, having a collocation database between RO events and convection over orographic regions is a valuable contribution, same as the detailed discussion of two cases

of WRF simulations. The comparison of RO events and WRF simulations, however, which is the core part of the paper, is incomplete and based on wrong assumptions.

Still, I think that the two selected cases are good examples for discussing several important aspects that have to be considered when comparing high resolution simulations with real observations. Therefore resubmission of a revised manuscript is encouraged.

If the manuscript were to be revised and resubmitted, several major issues have to be addressed.

**Major Concerns**

(I) Firstly, it is obvious that for both detailed comparisons, the Alps and Andes cases, the waves simulated with WRF are not seen in the corresponding RO soundings.

Following Alexander et al. (2008), the amplitude attenuation factor E can be calculated.

For the Alps case, E is $<10^{-11}$

For the Andes case, E is $<0.03$

Amplitudes in the RO soundings are 2K, sometimes more. For explaining these amplitudes with the WRF simulations, simulated amplitudes would have to be $>60$K. This is physically not reasonable.

In the Andes case the tangent points do not even hit the region of simulated wave trains. Further, the simulated wave trains are quite fuzzy, suggesting that amplitude attenuation should be even stronger. Details on the calculation of E are given below in a separate section.

(II) This clearly shows that the WRF simulations alone are not sufficient to explain the observations. In some cases they may be sufficient, but generally they are

not.

The simplest explanation is that horizontal wavelengths of the observed waves are usually much longer than those of the simulated waves. This possibility and its consequences are not discussed and completely ignored. The wavelet analysis even stops at 120km, which is at the verge of the range of waves that can be seen by GPS RO.

One explanation could be mountain waves and convective waves of longer scales that are not captured by the WRF simulations, but co-exist with the simulated short scale wave modes.

Another explanation could be **gravity waves emitted from jets and fronts**. Surprisingly, this third major source of gravity waves has been completely disregarded in the manuscript. Just as an example, in the beginning of the abstract only orography and convection are listed as major wave sources.

Usually gravity waves emitted from jets and fronts are of larger scale and could co-exist with the small scale waves that occur in the WRF simulations. Since the Alps case is in the winter season when gravity waves emitted from jets and fronts are quite abundant, ignoring this wave source is not possible.

(III) Thirdly, since the WRF simulations are insufficient to describe the observed waves, meteorological data sets, such as ECMWF analyses, could be investigated whether larger scale wave patterns are found that can explain the RO observations.

GPS RO measures temperatures. Therefore analysis of temperature fluctuations is preferable to the analysis of vertical wind w', also in the WRF simulations. Focusing too much on w' will bias any analysis towards high-frequency waves that are difficult for GPS RO to observe. Moreover, a direct comparison of amplitudes is not possible.

(IV) Fourthly, including a detailed discussion of vertical wavelength biases in the manuscript will distract the readers. For the examples given, observed and simulated waves are obviously different and wavelengths extracted from the simulations cannot be assumed for the observations. Therefore it is not clear whether these sampling biases apply.

**Amplitude attenuation factors for the Alps and Andes cases**

For GPS RO an amplitude attenuation factor E was deduced by Alexander et al. (2008)

$$E = \exp\left(-\frac{R_E}{2H}\tan^2(\gamma)\cos^2(\Theta)\right) \tag{1}$$

The aspect ratio $\alpha_h$ is the ratio of vertical to horizontal wavelength $\alpha_h = \lambda_z/\lambda_h$. The tilt angle of the wavefronts with respect to the horizontal surface is called $\gamma$, $\alpha_h = |\tan(\gamma)|$. The angle $\Theta$ is the angle between the LOS and the horizontal projection of the wave vector. $R_E = 6371km$ is the Earth radius, $H = 7km$ the scale height.

Citation: Alexander, P., A. de la Torre, and P. Llamedo (2008), Interpretation of gravity wave signatures in GPS radio occultations, J. Geophys. Res., 113, D16117, doi:10.1029/2007JD009390.

If the waves simulated with WRF are observable with GPS RO, the wave parameters taken from the simulations have to result in reasonable values of E well above 0.1, which is however not fulfilled.

**E in the Alps case**

From the manuscript, we read:
l.321: $\Theta$=0deg    l.220: $\lambda_h$=20 or 60km    l.283: $\lambda_z$=15km
These numbers are inserted into Eq. (1).

for $\lambda_h$=20km: $\tan\gamma$=15/20=0.75 $\rightarrow E = 7 \cdot 10^{-112}$

for $\lambda_h$=60km: $\tan\gamma$=15/60=0.25 $\rightarrow E = 4 \cdot 10^{-13}$

**E in the Andes case**

From the manuscript, we read:
l.397: $\Theta$=80deg   l.374, l375: $\lambda_h$=20 or 40km   l.382: $\lambda_z$=20 to 25km, 20km is used here

These numbers are inserted into Eq. (1).

for $\lambda_h$=20km: $\tan\gamma$=20/20=1 $\rightarrow E = 10^{-6}$

for $\lambda_h$=40km: $\tan\gamma$=20/40=0.5 $\rightarrow E = 0.03$

---

## Referee Comment (RC2) · Anonymous Referee #2 · 14 Jan 2018

Review for Hierro et al., "*Orographic and convective gravity waves above the Alps and Andes mountains during GPS radio occultation events – a case study*"

Summary:

This paper analyzed the collocated GPS radio occultation profiles near the convective systems identified from ISCCP over two orographic regions of Alps and Andes. Out of a total of 10 collocated RO profiles, one RO sounding (bending angle and temperature profiles) from each region was analyzed. The convective cloud top height was identified. Gravity wave (GW) analysis over both selected regions were also carried out. The gravity wave signature from the two case studies were derived base on the WRF simulation and the RO vertical profiles.

The GWs with horizontal wavelengths of 20 km and 40 km (two mode) and vertical wavelengths (15 km or 20-25 km) were identified from the WRF simulation over the two regions. The vertical propagating GWs with "apparent" vertical wavelength of ~4 km over Alps, or ~4.5 km over Andes were also identified from GPS RO soundings. The so-called distortion factor was also investigated, which measures the discrepancy between the "apparent" (or measured) and the "real" vertical wavelength affected by the RO observation geometry reference to the GWs propagation direction.

Overall, the paper use the case study to demonstrate the GPS RO capable of detecting the vertically transporting GW over the orographic regions, where mountain waves persistent as seen in the WRF simulations. The high vertical resolution GPS RO sounding has the advantage of detecting the fine vertical scale GWs but less capable of identify the GW with fine horizontal wavelength (e.g., Wu et al., 2006). The paper seems trying to synergize the GPS RO observation with the WRF simulation to present a more complete picture of the GW morphology, which has the merit for publication. However, it is not clearly stated in both the introduction and conclusion of the paper.

I would recommend publication of the paper after "major revision" with my comments below:

Major comments:

(1) The paper writing need some significant improvement to better describe the research work, including the grammar and sentence structures.
   a. Many super long sentences should be split into shorter sentences.
   b. The author(s) intend to use the "first" person throughout the paper. Generally the scientific paper should be written with third person.
(2) The motivation and key contribution of this paper need to clearly stated in the introduction. The authors add a short paragraph (L102-108) in the introduction, which was the details of work of the paper, but not "Why" to carry out such work. The motivation should be the science or technical questions that haven't been addressed, or what is new in this paper that advances the field of study.
(3) Only 5 collocated cases from each regions (Alps and Andes) were identified between GPS RO and ISCCP. And only one RO case for each region were used for the case studies. It is hard to believe that one single RO case is representative of such a large region,

especially hard to believe the single cloud-top-height identified from the bending/temperature anomaly method will be representative over such a large study domain.

  a.  The author's response shows 294 collocations over Alps but only 50 collocation over Andes within 3-hr and 100 km of the ISCCP convective systems, which should be included in the manuscript. The selection of the 10 case need to be justified.
  b.  Should consider demonstrate these RO soundings in **statistical sense** that they can detect consistently the gravity waves with vertical wavelength of 4-4.5 km, instead of randomly pick one out of 5 cases.
  c.  Also ISCCP website shows the DX (B3) data are available 07/01/1983 - 12/31/2009.
  d.  More robust analysis of the GW from more collocated GPS RO soundings near convective systems will strengthen the selection of the "representative" case for further WRF simulation study and improve the paper quality.

(4)  Some technical details are missing and need to be added,

  a.  What exact parameters from ISCCP were used?
  b.  How ISCCP identify the convective system? Any uncertainty related to the usage of infrared CTT to identify the convective system, especially over cold surface?
  c.  How to detect the CTH from the anomaly method (Biondi et al, 2012)? What threshold etc. used to identify the CTH in Fig. 1a,b? Can't just cite the paper. The basic details are still needed to be included in the paper.
  d.  Discussion on the result out of CWT is missing for both Fig. 5 and Fig. 9. What exactly plotted (e.g., parameters) need to be detailed described in the manuscript, and should be after the discussion of Fig. 4. The author(s) sometimes jump the discussion without following the orders of the figures.
  e.  Section 2: missing information of the duration of RO data used in this paper, e.g., what years data were used, are they same for different RO missions?
  f.  L156: need description of the reference climatology profile, e.g., how many year average, horizontal, vertical sampling interval, how to do average etc.
  g.  L159: WRF model description needed, e.g., version, horizontal, vertical resolution, citation.

(5)  Figures need to be improved:

  a.  Fig. 1: The four cases in smaller inlet are hardly legible.
      i.  Might consider combine all five (or more cases) into one statistical plot plus the one individual plot in the middle panel.
      ii.  RO profiles from which RO mission should be indicated in plot or the manuscript.
  b.  Fig. 8&9 should be consistent with Fig. 4&5, respectively
      i.  Why Fig. 8 only has three UTC time but Fig. 4 shows four UTC time?
      ii.  Why adding temperature wavelet analysis result in Fig. 9d but not in Fig. 5?
  c.  Fig. 10:

i. The distortion factor could explain the discrepancy of the GW vertical wavelength seen in GPS RO (4 – 4.5 km, L322, ) as compared to the WRF simulation (~15 km, L288, ~20-25km, L421). But why the two "green circle" at those certain locations were not discussed. Will the plot different for other region, e.g., over Alps?

(6) Most of the figure captions did not describe what is plotted and need to be updated.

   a. The reader should be able to understand the figure without the need to consult the text. It might be worth consulting the paper by de la Torre A. et al. (2006) or others on caption writing.

   b. All the color bars do not have "UNITs" either on plot or in the caption.

   c. Fig. 3: There are no description on what is plotted in Figure 3, variance??. You can't simply say it is "GW" structure.

   d. Same for Fig. 5, 6c, 7,8, 9c. The details in each panel need to be clearly described in caption succinctly.

Technical comments:

L40: "Vertical profiles … (Kursinski and Gebhardt, 2014). Need to be rewritten to be parallel statement.

L42: "0.1-0.3 g kg$^{-1}$"

L48: "troposphere and lower stratosphere

L72: "storm"

L74: "strongly affects"

L83: "Fovel et al., …, generating high-frequency???" Sentence is not complete.

L102 -108: Not a motivation, need to be rewritten.

L109: "Section 2…"

L125: RO data duration should be added

L130: Figure 1 caption should include what is plotted. E.g., the elevation map, Alps over Europe and Andes in South America…

L143: According

L146: "interval" → "difference"

L156: Missing description of the reference climatology profile, e.g., how many year average, horizontal, vertical sampling interval, how to do average etc.

L172: The total number of collocations (e.g, 294 over Alps, 50 over Andes) should be mentioned, and better to show the statistical results of the analysis instead of the hand-pick of the 5 cases. Or justification of the representativeness of the selected cases are required.

L180: Fig.1 need to be updated. Hardly legible.

L183: Caption should describe/mention each panel. Very hard to understand and need to be rewritten.

L195: 3.1. Case study over the Alps region

L205: What is plotted in Fig. 3 need to be clearly stated.

L240: They reach again… after a partial… → They reach large amplitudes again in stratosphere after … tropopause at around 11km.

L247: in red .

L250: (CWT) corresponding to Figs. 4a, b, and d, …,  Why missing Fig. 4c??

L311: "perfect" band-pass, What does that mean?

L297: Fig. 6c caption should describe what is plotted, basically what is the results after CWT, plus the unit of the color bar.

L313: "This method has two steps: …, …, and to force a zero mean." Super long sentence and need to be broken up into smaller pieces.

L316: We keep in mind that → Note that.

L328: prevent us  (from) observing

L371: 3.2. Case study over the Andes region

L377: What does "[7;-3]" mean? It was shown in many places.

L390: caption need to describe what is plotted, plus the unit.

L414: Missing detailed description and discussion of Fig. 9.

L415: Fig. 8, Zonal variation of w in WRF simulation at ??altitude in three UTC times (13, 17, and 21 UTC) over the Andes region. The GPS RO took place at 16:56 UTC with mean LTP at ….

L440: Similar to Fig. 6, but for the RO case study from Fig. 1b over the Andes region.

L455: Equation (1): should $\cot(\alpha)$ be $\cos(\alpha)$ instead?

L465: please describe why the two green circles at the specific "propagation angle".

L474: were → where

---

## Author Comment (AC1) · 19 Feb 2018

**Response to Referee #1**

**"Orographic and convective gravity waves above the Alps and Andes mountains during GPS radio occultation events — a case study" by Rodrigo Hierro et al.**

We acknowledge very much the detailed comments and suggestions made by Referee 1.

General Comment
**In my AMTD access review several major concerns were raised which showed that a major rewriting and refocusing of the manuscript would be required, probably exceeding the time usually allotted for revisions of a paper under review. Therefore my recommendation at this stage was to reject the paper. Since then, only minor changes were made. This is why this recommendation is repeated. Of course, having a collocation database between RO events and convection over orographic regions is a valuable contribution, same as the detailed discussion of two cases of WRF simulations. The comparison of RO events and WRF simulations, however, which is the core part of the paper, is incomplete and based on wrong assumptions. Still, I think that the two selected cases are good examples for discussing several important aspects that have to be considered when comparing high resolution simulations with real observations. Therefore resubmission of a revised manuscript is encouraged. If the manuscript were to be revised and resubmitted, several major issues have to be addressed.**

**Major Concerns**
**(I) Firstly, it is obvious that for both detailed comparisons, the Alps and Andes cases, the waves simulated with WRF are not seen in the corresponding RO soundings. Following Alexander et al. (2008), the amplitude attenuation factor E can be calculated.**
**For the Alps case, E is $<10^{-11}$**
**For the Andes case, E is $<0.03$**
**Amplitudes in the RO soundings are 2K, sometimes more. For explaining these amplitudes with the WRF simulations, simulated amplitudes would have to be $>60K$. This is physically not reasonable. In the Andes case the tangent points do not even hit the region of simulated wave trains. Further, the simulated wave trains are quite fuzzy, suggesting that amplitude attenuation should be even stronger. Details on the calculation of E are given below in a separate section.**

Our revised manuscript, as it may be appreciated in the attached, lengthy new version "with track changes", has been completely rewritten. Old figures 4, 5, 8 and 9 were eliminated and replaced by the new figures 2, 5, 7, 8, 9 (new version). Old Figure 2 was enlarged in two plates (new figure 3). Many additional paragraphs are now included. The discussion regarding the expected wavelengths distortion was separately explained in an Appendix. New calculations and simulations regarding the 2 case studies, now including simultaneously WRF simulations, reanalysis data and RO measurements are included.

In particular, in base of this last reviewer's comment, a detailed calculation of the attenuation factor has been made. This factor has been obtained for the two case studies, based on the mathematical details provided in our paper Alexander et al. (2008). We analyzed the possibility of RO detection of 4 clearly different structures, at tropospheric

and stratospheric altitudes, during both case studies. These were not completely reported in our previous version. The results are shown in Table 1, now included. As it may be appreciated, 3 structures are expected to be, at least partially, detected. This is not possible for the fourth structure. For example, in the Andes case study, we found that different segments of the same profile exhibit GW belonging to different sources.

Please notice that the new simulations and our new interpretations of the corresponding GW lead to different attenuation values, as aspect ratios and angles between LOS and GW fronts have now changed.

**(II) This clearly shows that the WRF simulations alone are not sufficient to explain the observations. In some cases they may be sufficient, but generally they are not. The simplest explanation is that horizontal wavelengths of the observed waves are usually much longer than those of the simulated waves. This possibility and its consequences are not discussed and completely ignored. The wavelet analysis even stops at 120km, which is at the verge of the range of waves that can be seen by GPS RO. One explanation could be mountain waves and convective waves of longer scales that are not captured by the WRF simulations, but co-exist with the simulated short scale wave modes.**

The analysis now includes ERA Interim reanalyses data and an extensive discussion is given about the need to consider this information to perform a reliable study of the observed GW structures, mainly for the Alps case study (Sec 3.1.1.to 3.1.3. and 3.2.1.) The wavelet analysis is now applied only to the RO $\delta T$ profiles at both regions (3.1.2. and 3.2.1.)

**Another explanation could be gravity waves emitted from jets and fronts. Surprisingly, this third major source of gravity waves has been completely disregarded in the manuscript. Just as an example, in the beginning of the abstract only orography and convection are listed as major wave sources. Usually gravity waves emitted from jets and fronts are of larger scale and could co-exist with the small scale waves that occur in the WRF simulations. Since the Alps case is in the winter season when gravity waves emitted from jets and fronts are quite abundant, ignoring this wave source is not possible.**

We completely agree with this comment. Here, we carefully selected our case studies, taking into account the convenience of considering only scenarios in the absence of jets and fronts. In doing so, we would expectedly filter out GW from at least one or two of these major sources, thus avoiding an extremely intricate analysis, perhaps not possible at all.

**(III) Thirdly, since the WRF simulations are insufficient to describe the observed waves, meteorological data sets, such as ECMWF analyses, could be investigated whether larger scale wave patterns are found that can explain the RO observations. GPS RO measures temperatures. Therefore analysis of temperature fluctuations is preferable to the analysis of vertical wind w', also in the WRF simulations. Focusing too much on w' will bias any analysis towards high-frequency waves that are difficult for GPS RO to observe. Moreover, a direct comparison of amplitudes is not possible.**

In the new version, in addition to the included reanalyses data, we restricted ourselves to show vertical velocity simulations only in relation with mountain waves. The inclusion of $\delta T$ from simulations and reanalysis data is now considered and shown in all the new figures and results.

**(IV) Fourthly, including a detailed discussion of vertical wavelength biases in the manuscript will distract the readers. For the examples given, observed and simulated waves are obviously different and wavelengths extracted from the simulations cannot be assumed for the observations. Therefore it is not clear whether these sampling biases apply.**

Two of our main changes in the manuscript are the inclusion of the expected amplitude attenuation and wavelength distortion during the collocated RO measured profiles. This last discussion is now moved to the Appendix. We feel that now it will not distract from the main objectives of the manuscript.

---

## Author Comment (AC2) · 19 Feb 2018

**Response to Referee #2**

**"Orographic and convective gravity waves above the Alps and Andes mountains during GPS radio occultation events — a case study" by Rodrigo Hierro et al.**

We acknowledge very much the detailed comments and suggestions made by Referee 2.

**Summary: This paper analyzed the collocated GPS radio occultation profiles near the convective systems identified from ISCCP over two orographic regions of Alps and Andes. Out of a total of 10 collocated RO profiles, one RO sounding (bending angle and temperature profiles) from each region was analyzed. The convective cloud top height was identified. Gravity wave (GW) analysis over both selected regions were also carried out. The gravity wave signature from the two case studies were derived base on the WRF simulation and the RO vertical profiles. The GWs with horizontal wavelengths of 20 km and 40 km (two mode) and vertical wavelengths (15 km or 20-25 km) were identified from the WRF simulation over the two regions. The vertical propagating GWs with "apparent" vertical wavelength of ~4 km over Alps, or ~4.5 km over Andes were also identified from GPS RO soundings. The so-called distortion factor was also investigated, which measures the discrepancy between the "apparent" (or measured) and the "real" vertical wavelength affected by the RO observation geometry reference to the GWs propagation direction. Overall, the paper use the case study to demonstrate the GPS RO capable of detecting the vertically transporting GW over the orographic regions, where mountain waves persistent as seen in the WRF simulations. The high vertical resolution GPS RO sounding has the advantage of detecting the fine vertical scale GWs but less capable of identify the GW with fine horizontal wavelength (e.g., Wu et al., 2006). The paper seems trying to synergize the GPS RO observation with the WRF simulation to present a more complete picture of the GW morphology, which has the merit for publication. However, it is not clearly stated in both the introduction and conclusion of the paper. I would recommend publication of the paper after "major revision" with my comments below:**

**Major comments:**

**(1) The paper writing need some significant improvement to better describe the research work, including the grammar and sentence structures.**

**a. Many super long sentences should be split into shorter sentences.**
**b. The author(s) intend to use the "first" person throughout the paper. Generally the scientific paper should be written with third person.**

Our revised manuscript, as it may be appreciated in the attached, lengthy new version "with track changes", has been completely rewritten. Along these track changes, the general reduction of sentences as well as the use of third person may be appreciated. The grammar was revised by an expert. Old figures 4, 5, 8 and 9 were eliminated and replaced by the new figures 2, 5, 7, 8, 9 (new version). Old Figure 2 was enlarged (new figure 3). Many additional paragraphs are now included. The discussion regarding the expected wavelengths distortion was separately explained in an Appendix. New

calculations and simulations regarding the 2 case studies, now including simultaneously WRF simulations, reanalyses data and RO measurements are included.

**(2) The motivation and key contribution of this paper need to clearly stated in the introduction. The authors add a short paragraph (L102-108) in the introduction, which was the details of work of the paper, but not "Why" to carry out such work. The motivation should be the science or technical questions that haven't been addressed, or what is new in this paper that advances the field of study.**

We now feel that all of these comments and suggestions have been fulfilled in our new version (please see Sec 1).

**(3) Only 5 collocated cases from each regions (Alps and Andes) were identified between GPS RO and ISCCP. And only one RO case for each region were used for the case studies. It is hard to believe that one single RO case is representative of such a large region, especially hard to believe the single cloud-top-height identified from the bending/temperature anomaly method will be representative over such a large study domain.**
 **a. The author's response shows 294 collocations over Alps but only 50 collocation over Andes within 3-hr and 100 km of the ISCCP convective systems, which should be included in the manuscript. The selection of the 10 case need to be justified.**
**b. Should consider demonstrate these RO soundings in statistical sense that they can detect consistently the gravity waves with vertical wavelength of 4-4.5 km, instead of randomly pick one out of 5 cases.**

As we now explain in the text (line 186-7): "In the present study, a GW climatology from the available limited number of collocated cases is not intended". The response to these two comments (a. and b.), as well as the pre-selection and later selection of case studies are detailed and discussed in lines 187-197.

**c. Also ISCCP website shows the DX (B3) data are available 07/01/1983 - 12/31/2009.**

This point is now explained in several segments of text between lines 140-171

**d. More robust analysis of the GW from more collocated GPS RO soundings near convective systems will strengthen the selection of the "representative" case for further WRF simulation study and improve the paper quality.**

Our final objective in this paper was only a detailed analysis of two cases study, however we fully agree with this comment.

**(4) Some technical details are missing and need to be added,**
**a. What exact parameters from ISCCP were used?**

The parameters extracted from ISCCP data, are: time of occurrence, center (mass center) and radius of the storm. The COSMIC mission started in June 2006.

**b. How ISCCP identify the convective system? Any uncertainty related to the usage of infrared CTT to identify the convective system, especially over cold surface?**

A discussion detailing all the information available for us regarding the used cloud data is provided in Sec. 2. We used time of occurrence, the center of the storm (mass center) and radius of the storm. We don't have access to ISCCP methodology so we just used their database.

**c. How to detect the CTH from the anomaly method (Biondi et al, 2012)? What threshold etc. used to identify the CTH in Fig. 1a,b? Can't just cite the paper. The basic details are still needed to be included in the paper.**

Here we are just able to answer in a similar way as before, regarding ISCCP. We don't have access to the methodology used in Biondi et al. (2012). On the other hand, in the mentioned paper, details of the procedure used by the authors to detect CTH, are available.

**d. Discussion on the result out of CWT is missing for both Fig. 5 and Fig. 9. What exactly plotted (e.g., parameters) need to be detailed described in the manuscript, and should be after the discussion of Fig. 4. The author(s) sometimes jump the discussion without following the orders of the figures.**

We agree with this observation. The original text has been fully rewritten and we feel that a new order is given to the description of parameters, figures and hypotheses. Previous (some of them modified) and new results may be now appreciated. In particular, the CWT analysis is now limited only to both GPS RO $\delta T$ profiles.

**e. Section 2: missing information of the duration of RO data used in this paper, e.g., what years data were used, are they same for different RO missions?**

This information is now included in the text.

**f. L156: need description of the reference climatology profile, e.g., how many year average, horizontal, vertical sampling interval, how to do average etc.**

**g. L159: WRF model description needed, e.g., version, horizontal, vertical resolution, citation.**

Please see lines 172-186

**(5) Figures need to be improved:**
**a. Fig. 1: The four cases in smaller inlet are hardly legible.**
**i. Might consider combine all five (or more cases) into one statistical plot plus the one individual plot in the middle panel.**
**ii. RO profiles from which RO mission should be indicated in plot or the manuscript.**
**b. Fig. 8&9 should be consistent with Fig. 4&5, respectively**
**i. Why Fig. 8 only has three UTC time but Fig. 4 shows four UTC time?**
**ii. Why adding temperature wavelet analysis result in Fig. 9d but not in Fig. 5?**

As mentioned above, old figures 4, 5, 8 and 9 were eliminated and replaced by the new figures 2, 5, 7, 8, 9. The plates in the new figure 3 are enlarged. We feel that the results are more clear and consistent now.

**c. Fig. 10:**
**i. The distortion factor could explain the discrepancy of the GW vertical wavelength seen in GPS RO (4 – 4.5 km, L322, ) as compared to the WRF simulation (~15 km, L288, ~20-25km, L421). But why the two "green circle" at those certain locations were not discussed. Will the plot different for other region, e.g., over Alps?**

Now, part of the distortion discussion was moved to an appendix. Other reviewer suggested to not distract the reader with these considerations too much. Nevertheless, the two geen circles, suggesting possible under- and/or overestimations for a given set of parameters is included for illustration only.

**(6) Most of the figure captions did not describe what is plotted and need to be updated.**
**a. The reader should be able to understand the figure without the need to consult the text. It might be worth consulting the paper by de la Torre A. et al. (2006) or others on caption writing.**
**b. All the color bars do not have "UNITs" either on plot or in the caption.**

All of these points have been carefully taken into account in the preparation of the new version. We hope that the descriptions are more clear now.

**c. Fig. 3: There are no description on what is plotted in Figure 3, variance??. You can't simply say it is "GW" structure.**

The GW perturbation to the vertical velocity is explicitly written in the caption and in the text.

**d. Same for Fig. 5, 6c, 7,8, 9c. The details in each panel need to be clearly described in caption succinctly.**

Here we apply the same comment as the two before.

**Technical comments:**
**L40: "Vertical profiles … (Kursinski and Gebhardt, 2014). Need to be rewritten to be parallel statement.**
Done.

**L42: "0.1-0.3 g kg-1 "**
Done.

**L48: "troposphere and lower stratosphere**
Done.

**L72: "storms"**

Done.

**L74: "strongly affects"**
Done.

**L83: "Fovel et al., …, generating high-frequency???" Sentence is not complete.**
Done.

**L102 -108: Not a motivation, need to be rewritten.**
Done.

**L109: "Section 2…"**
Done.

**L125: RO data duration should be added**
Done.

**L130: Figure 1 caption should include what is plotted. E.g., the elevation map, Alps over Europe and Andes in South America…**
Done.

**L143: According**
Done.

**L146: "interval" à "difference"**
Done.

**L156: Missing description of the reference climatology profile, e.g., how many year average, horizontal, vertical sampling interval, how to do average etc.**
We used the data from the climatology, as provided by the Wegener Center. Nevertheless, some details regarding how it was obtained were not available for us.

**L172: The total number of collocations (e.g, 294 over Alps, 50 over Andes) should be mentioned, and better to show the statistical results of the analysis instead of the handpick of the 5 cases. Or justification of the representativeness of the selected cases are required.**
An detailed explanation about this comment is included in the text now.

**L180: Fig.1 need to be updated. Hardly legible.**
Done.

**L183: Caption should describe/mention each panel. Very hard to understand and need to be rewritten.**
This was considered in all the new and old figures.

**L195: 3.1. Case study over the Alps region**
Done.

**L205: What is plotted in Fig. 3 need to be clearly stated.**
Done.

**L240: They reach again… after a partial… à They reach large amplitudes again in stratosphere after … tropopause at around 11km.**
New figures, corresponding captions and discussions in the text are now included.

**L247: in red for altitude reference. L250: (CWT) corresponding to Figs. 4a, b, and d, …, Why missing Fig. 4c??**
This figure is now removed.

**L311: "perfect" band-pass, What does that mean?**
A comment is included at lines 284-285.

**L297: Fig. 6c caption should describe what is plotted, basically what is the results after CWT, plus the unit of the color bar.**
A text is now included in the figure caption.

**L313: "This method has two steps: …, …, and to force a zero mean." Super long sentence and need to be broken up into smaller pieces.**
Done.

**L316: We keep in mind that à Note that.**
Done.

**L328: prevent us to (from) observing**
Done.

**L371: 3.2. Case study over the Andes region**
Done.

**L377: What does "[7;-3]" mean? It was shown in many places.**
The horizontal mean wind components, now explained in the text.

**L390: caption need to describe what is plotted, plus the unit.**
Figure now removed.

**L414: Missing detailed description and discussion of Fig. 9.**
This discussion was moved to an Appendix and partially referred to a very recent accepted paper, following a suggestion of another reviewer.

**L415: Fig. 8, Zonal variation of w in WRF simulation at ??altitude in three UTC times (13, 17, and 21 UTC) over the Andes region. The GPS RO took place at 16:56 UTC with mean LTP at ….**
This figure is now removed.

**L440: Similar to Fig. 6, but for the RO case study from Fig. 1b over the Andes region.**
Done.

**L455: Equation (1): should cot(a) be cos(a) instead?**
It is correct as it is written (please see A18).

**L465: please describe why the two green circles at the specific "propagation angle".**

Because the $\alpha$ value represent the inclination of the LTP respect to the horizontal plane.

**L474: were → where**

Done.

---

## Author Response (AR2)

**Response to the reviewer:**

*We truly appreciate the comments and suggestions made by the reviewer.*

Review of the revised manuscript "Orographic and convective gravity waves above the Alps and Andes mountains during GPS radio occultation events – a case study" by Hierro et al.

The authors did a great job in addressing my major comments - congratulations!
The following revisions were made:
(I) The amplitude attenuation factor has been considered for the case studies discussed.
(II) It has been considered that WRF simulations will not always be sufficient to describe the waves seen in RO soundings.
(III) ERA Interim data were analyzed for the case in which WRF simulations were not sufficient to explain the RO soundings. For matching the RO soundings now also temperature fluctuations are considered, in addition to fluctuations of vertical wind
(IV) Discussion of wavelength biases was moved into an appendix.

Overall, the manuscript is much improved now and can be recommended for publication in AMT after addressing a couple of remaining minor, but important comments.

My main comments are:
* it is difficult to generally rule out the co-existence of jet-generated GWs in the subtropics, even for selected cases
* the text should be screened for consistency and some additional information added, some points are listed in my Minor Comments

In the following, line numbers refer to the manuscript version with changes NOT highlighted.

Minor comments:

(1) The existence of jet-generated GWs cannot be generally ruled out because your work focuses on the subtropics, and there the subtropical jets are more or less omnipresent. This is confirmed for your Alps and Andes cases in Figs. 5 and 9 where wind speeds of well above 20m/s are evidently seen in the troposphere.

Therefore my questions, concerning l.193, but also elsewhere:
What is the criterion you are using to make sure that GWs generated by jets or fronts are absent?

Are you sure that this criterion will always work?
The wave seen in ERA Interim (Fig.7) that could not be simulated by WRF could be a jet-generated GW.

*All changes made in the text of the revised manuscript are highlighted in yellow.*

*An additional paragraph and 3 references were included at lines 194-204 and a sentence in the Conclusions (lines 463-465). The systematic negative results regarding the existence of a possible unbalance of the flow were not included, only mentioned. For example, the cross-stream component of the Lagrangian Rossby number at 250 hPa in the Alps region, expected to be greater than 0.5 for unbalanced conditions, looks as follows:*

[Figure]

(2) l.318:
The MW should be considerably attenuated below AND ABOVE the tropopause, and therefore be invisible for RO soundings.
This is also what you discuss in Sect. 3.1.3.

*This comment was included at line 331.*

(3) l.329:
It should be mentioned that ERA Interim will give some information about longer horizontal wavelength GWs, but has a relatively coarse resolution and will strongly underestimate wave amplitudes. Accordingly, amplitudes in Fig. 7 are quite low. If this wave is seen in RO soundings, it would have a much larger amplitude.
Of course, higher resolution ECMWF data would have been better suited, however, they are probably difficult to obtain. Therefore I really appreciate the authors' efforts based on ERA Interim.

*A paragraph was included at lines 141-143.*

(4) After l.339: caption of Table 1
Please mention that line 1 in the table refers to the wave seen in ERA-Interim, while lines 2-4 in this table refer to the WRF simulations.

*This point was included at lines 358-359.*

(5) l.391-395: I guess, erroneously a wrong word was used in l.391!
Mountain waves cannot produce circular wave patterns! MWs usually have wave fronts that are aligned parallel to the mountain range, or, in case of isolated mountains, MWs can display ship-wave patterns.
Circular wave patterns are usually found above convective GW sources.

Suggestion: just change "orographic" in l.391 to "non-orographic"

*We agree with this statement, and it is commented at lines 402 and 404.*

(6) l.442: It should be mentioned that attenuation factors close to 1 have to be taken with some caution because "ideal" wave patterns are assumed for this calculation.

*This comment was included at lines 450-451.*

(7) l.499: below the tropopause -> below and above the tropopause

*Done.*

(8) l.500: delete "in the troposphere" and refer to the GWs simulated with WRF
suggestion:

attenuation factor in the troposphere confirms that the GW
->
attenuation factor confirms that these GW

*Done.*

(9) l.505: same problem as in l.391-395
orographic -> non-orographic

*Done.*

Other comments:

(1) l.28 In this last case -> In the Andes case

*Done.*

(2) l.43 less than -> better than

*Done.*

(3) l.90 determined -> showed

*Done.*

(4) l.104 intrinsic period -> intrinsic frequency

*Done.*

(5) l.110 anomay -> anomaly

*Done.*

(6) l.114/115
to Alps and Andes ranges.
->
to the Alps and Andes mountain ranges.

*Done.*

(7) l.185 inner one. -> inner ones.

*Done.*

(8) Fig.5: color scales for dw and dT are missing
In the figure caption it should be better clarified that in (a) dw is shown, while in (b) and
(c) dT is shown. The wording "...dw and dT..." with dw and dT so close together is
somehow misleading.

*Done.*

(9) l.323:
presence of GW with scale long enough is not able to be
->
presence of large scale GWs that are not able to be

*Done.*

(10) l.325: are observed. -> are analyzed.

*Done.*

(11) l.325: a defined -> a well defined

*Done.*

(12) l.327: in 0.90. -> to 0.90 (first line in Table 1).

*Done.*

(13) l.371 respectedly, -> respectively,

*Done.*

(14) l.376: "The TP altitudes." not a complete sentence!

*It was removed.*

(15) Caption of Fig.9, same problem with dw and dT as for Fig.5

*Done.*

(16) Text after l.415: These sentences are somehow twisted, please rewrite!
Suggestion:

Next, the wavelike structure of the RO T profile, retrieved at the Andes region. This profile is shown in the central panels of Fig 3b. Its horizontally projected LTP is seen in Fig. 8, is analyzed.
->
Next, the wavelike structure of the RO T profile retrieved at the Andes region is analyzed. This profile is shown in the central panels of Fig 3b, and its horizontally projected LTP is seen in Fig. 8.

*Done.*

(17) After l.420 in the revised manuscript, there is a dashed line. Possibly, a file conversion error. Please check!

*Done.*

(18) After l.450 in the revised manuscript:
Please check! It looks like an equation should have been deleted, but wasn't.

*Done.*

(19) l.475: orographic forcing and -> orographically forced or

*Done.*

(20) l.512: decreasing Lz. -> at the same time Lz decreases.

*Done.*

(21) l.518
A partial attenuation reveals
->
An evaluation of the partial attenuation coefficient reveals

*Done.*

(22) l.533:
Apendix -> Appendix

*Done.*
(23) l.542: This sentence is somehow twisted, suggestion:
It is defined the distortion as the ratio:

->
We define the distortion as the ratio:

*Done.*

(24) l.586: wasinitiated -> was initiated

*Done.*

(25) l.589: wassupported -> was supported

*Done.*

->
We define the distortion as the ratio: